# EFFICIENT PHYSICS-CONSTRAINED DIFFUSION MODELS FOR SOLVING INVERSE PROBLEMS

## ABSTRACT

Solving inverse problems in scientific and engineering domains often involves complex, nonlinear forward physics and ill-posed conditions. Recent advancements in diffusion model have shown promise for general inverse problems, yet their application to scientific domains remains less explored and is hindered by the complexity and high non-linearity of physics constraints. We present a physics-constrained diffusion model (PCDM) designed to solve inverse problems in scientific and engineering domains by efficiently integrating pre-trained diffusion models and physics-constrained objectives. We leverage accelerated diffusion sampling to enable a practical generation process while strictly adhering to physics constraints by solving optimization problems at each timestep. By decoupling the likelihood optimization from the reverse diffusion steps, we ensure that the solutions remain physically consistent, even when employing fewer sampling steps. We validate our method on a wide range of challenging physics-constrained inverse problems, including data assimilation, topology optimization, and full-waveform inversion. Experimental results show that our approach significantly outperforms existing methods in efficiency and precision, making it practical for real-world applications.

## 1 INTRODUCTION

Inverse problems arise from various scientific and engineering fields, such as computational imaging (Beck & Teboulle, 2009), data assimilation (Evensen, 1994), optimal design (Bendsøe & Kikuchi, 1988), and geophysics (Tarantola, 1984). The goal of solving an inverse problem is to recover underlying data or physical properties $\boldsymbol{x} \in \mathbb{R}^n$ from observed measurements $\boldsymbol{y} \in \mathbb{R}^m$,

$$\boldsymbol{y} = \mathcal{A}(\boldsymbol{x}) + n, \tag{1}$$

where $\mathcal{A} : \mathbb{R}^n \to \mathbb{R}^m$ is the physical forward operator, and $n \in \mathbb{R}^m$ is additive noise. The physical forward operators often involve sophisticated simulations governed by partial differential equations (PDEs) or physics constraints, incorporating years of domain knowledge, and occasionally involve measurement operators. These inverse problems are typically ill-posed where multiple possible solutions $\boldsymbol{x}$ exist for a measurement $\boldsymbol{y}$. To deal with this challenge, a common approach is to solve the physics-constrained optimization problem with regularization reflecting the prior or underlying structural information of the solution,

$$\min_{\boldsymbol{x}} \frac{1}{2}\|\boldsymbol{y} - \mathcal{A}(\boldsymbol{x})\|_2^2 + \lambda\mathcal{R}(\boldsymbol{x}), \tag{2}$$

where $\mathcal{L}(\boldsymbol{x}) = \frac{1}{2}\|\boldsymbol{y} - \mathcal{A}(\boldsymbol{x})\|_2^2$ is an objective function that stems from the likelihood of alignment for the physics constraints, $\mathcal{R}(\boldsymbol{x})$ is the regularization on the $\boldsymbol{x}$, and $\lambda$ is the weight coefficient. Traditional choices for $\mathcal{R}$ using hand-crafted priors (Rudin et al., 1992) are not expressive enough to capture complicated data structures. With the rise of deep learning, the trends have shifted toward using deep generative models (Kingma, 2013; Goodfellow et al., 2014) as learned priors, which are more effective in representing intricate data structure (Ulyanov et al., 2018; Mosser et al., 2020; Patel & Oberai, 2021; Jacobsen & Duraisamy, 2022; Meng et al., 2022; Patel et al., 2022).

In recent years, diffusion models have demonstrated remarkable success in generative modeling of underlying data distributions and showed outperforming existing generative models in sampling

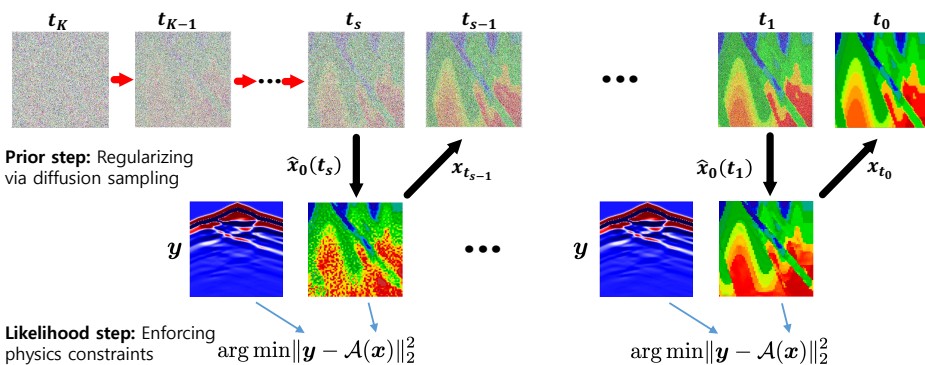

Figure 1: A schematic diagram of the physics-constrained diffusion models (PCDM). PCDM leverages accelerated diffusion sampling (prior step) to enable the reverse process in fewer timesteps ($K < T$), while strictly enforcing physics constraints by solving optimization problems (likelihood step) at each timesteps.

quality and stable training Ho et al. (2020); Song et al. (2021a;b); Dhariwal & Nichol (2021). Inspired by these results, incorporating pre-trained diffusion priors and likelihood gradients of alignment with the measurements to estimate the posterior score for solving the inverse problems within a Bayesian framework (Song et al., 2022; Chung et al., 2022; 2023; Song et al., 2023; 2024; Chung et al., 2024; Li et al., 2024). Specifically, in the inverse problems within the scientific and engineering domains, interest has grown in these line of methods for applications such as full-waveform inversion (FWI) (Wang et al., 2023; 2024; Taufik et al., 2024), topology optimization (TO) (Mazé & Ahmed, 2023; Giannone et al., 2023; Bastek et al., 2024), and data assimilation (or restoring missing data) (Shu et al., 2023; Jacobsen et al., 2023; Rozet & Louppe, 2023; Huang et al., 2024). However, the forward operator $\mathcal{A}$ in these domains comes in diverse and often incurs huge bottlenecks to solve due to its significant computational complexity and high nonlinearity. Existing methods could lead to slow inference times when naively incorporating a gradient update of the likelihood into every reverse sampling process of diffusion models, or suboptimal results if the total number of likelihood gradient update is not enough to fully optimize the loss $\mathcal{L}(\boldsymbol{x}) = \frac{1}{2}\|\boldsymbol{y} - \mathcal{A}(\boldsymbol{x})\|_2^2$, particularly when using accelerated sampling methods such as denoising diffusion implicit models (DDIM) (Song et al., 2021a). Therefore, it is necessary to obtain plausible solutions that strictly adhere to the constraints within a feasible time for these physics-constrained inverse problems.

To address these issues, we propose a physics-constrained diffusion model (PCDM) that treats the prior and likelihood steps separately and enhances their efficiency without affecting each other. During inference, we utilize accelerated diffusion models, such as DDIM, to update the prior steps at the subsets of the full-time trajectory. Following each prior step, we optimize the physics-constrained objective $\|\boldsymbol{y} - \mathcal{A}(\boldsymbol{x})\|_2^2$, allowing multi-step minimization, starting from denoised estimates from the prior steps as the initial guess, ensuring that the solution strictly adheres to the constraints. Our approach is particularly suited for physics-constrained inverse problems that are notoriously complex and highly nonlinear, making it practical and feasible. Finally, we evaluate our method across a wide range of physics-constrained inverse problems, including data assimilation, topology optimization, and full-waveform inversion. Compared to existing methods, our approach demonstrates superior performance in terms of accuracy and speed. The key contributions of our work are as follows:

- We present a physics-constrained diffusion model (PCDM) designed to address inverse problems in the scientific and engineering domains by efficiently integrating physical knowledge with diffusion models.

- Our method offers a feasible generation process using accelerated diffusion sampling, while strictly adhering to the physics constraints by solving optimization problems at fewer timesteps.

- From pre-training diffusion models from scratch for each physics-constrained problem benchmark, we demonstrated that our model outperforms existing approaches in terms of accuracy within a feasible time and is readily applicable to a variety of physical problems.

## 2 RELATED WORKS

In recent years, machine learning has been emerging in solving inverse problems in scientific and engineering domains. Roughly speaking, its applications can be categorized into two main approaches. The first category is supervised end-to-end methods (Li et al., 2021; Lu et al., 2021), which directly learn the inverse mapping from paired inputs and outputs. These methods enable fast evaluations and rely solely on observational data without accessing the physical solvers. For these practical advantages, it has been widely used for physics-related inversion problems (Wu & Lin, 2020; Zhang & Lin, 2020; Nie et al., 2021; Wang et al., 2022; Molinaro et al., 2023). However, due to operating in a supervised manner, they typically require thousands of paired data generated by physics-based solvers in advance and are often limited in the zero-shot scenarios where the observation process differs from the training conditions. Additionally, because these methods do not explicitly enforce physical constraints during inference, they can produce unrealistic outputs that violate the underlying physics, leading to significant issues.

The second category involves unsupervised (or semi-supervised) plug-and-play methods which typically leverage deep generative models as learned priors (Chung et al., 2023; Zhu et al., 2023; Mardani et al., 2024; Zhang et al., 2024; Wu et al., 2024). These methods do not rely on large amounts of paired data and instead focus on capturing the underlying distribution of plausible solutions in an unsupervised manner. Then, the pre-trained generative prior serves as a regularizer to generate plausible solutions when solving physics-constrained optimization problems. By doing so, they allow for more flexible and data-efficient solutions to inverse problems, particularly in cases where collecting paired data is expensive. For these advantages, it has been emerging for inversion problems in scientific and engineering domains (Raissi et al., 2019; Karniadakis et al., 2021; Mazé & Ahmed, 2023; Shu et al., 2023; Jacobsen et al., 2023; Rozet & Louppe, 2023; Huang et al., 2024; Wang et al., 2023; 2024; Taufik et al., 2024). While the methods can provide flexibility and robustness in dealing with physics-constrained inverse problems, the reliance on iterative solvers can result in slower inference times compared to supervised end-to-end methods. Nevertheless, the combinations of learned priors and physics-based constraints hold great promise for addressing the limitations of pure data-driven approaches. We focus on utilizing promising generative models, such as diffusion models, and improving the efficiency of the optimization process to make this method practical.

## 3 METHODS

### 3.1 PRELIMINARIES: DIFFUSION MODELS

Diffusion models (DM) (Ho et al., 2020; Song et al., 2021b) is an emerging generative model that employs both forward and reverse processes to learn the unknown data distribution progressively. In the forward process, clean data $\boldsymbol{x}_0 \in \mathbb{R}^n$ is drawn from an unknown data distribution $\boldsymbol{x}_0 \sim p_0$, DM progressively diffuses the data towards tractable distribution, such as Gaussian distribution with the following forward stochastic differential equations (SDE):

$$d\boldsymbol{x}_t = -\frac{\beta_t}{2}\boldsymbol{x}_t dt + \sqrt{\beta_t}d\boldsymbol{w}, \tag{3}$$

where $\beta_t$ is the noise schedule, and $w \in \mathbb{R}^n$ is the Wiener process at time $t \in [0, T]$. Here, $\beta_t$ typically increases monotonically with time, ensuring that for sufficiently large time steps, the distribution $\boldsymbol{x}_T \sim p_T$ approaches some prior distribution or Gaussian at the terminal time $T$. The reverse of this process is described by

$$d\boldsymbol{x}_t = \left[ -\frac{\beta_t}{2}\boldsymbol{x}_t - \beta_t \nabla_{\boldsymbol{x}_t} \log p_t(\boldsymbol{x}_t) \right] dt + \sqrt{\beta_t}d\bar{\boldsymbol{w}}, \tag{4}$$

where $p_t$ denotes the marginal density of $\boldsymbol{x}_t$ at time $t$, and $\nabla_{\boldsymbol{x}_t} \log p_t(\boldsymbol{x}_t)$ represents the score function (Song et al., 2021b). In practice, the score $\nabla_{x_t} \log p_t(\boldsymbol{x}_t)$ can be learned using a score network $\boldsymbol{s}_{\boldsymbol{\theta}}(\boldsymbol{x}_t, t)$ trained with the denoising score matching objectives (Vincent, 2011)

$$\min_{\theta} E_{\boldsymbol{x}_0, t, \boldsymbol{x}_t \sim p(\boldsymbol{x}_t|\boldsymbol{x}_0)} \left[ \left\| \boldsymbol{s}_{\boldsymbol{\theta}}(\boldsymbol{x}_t, t) - \nabla_{\boldsymbol{x}_t} \log p_t(\boldsymbol{x}_t|\boldsymbol{x}_0) \right\|_2^2 \right]. \tag{5}$$

It has also been shown that the denoising score matching is equivalent to the epsilon matching, as given by the relation, $\boldsymbol{s}_{\boldsymbol{\theta}}(\boldsymbol{x}_t, t) \approx -\frac{\boldsymbol{\epsilon}_{\boldsymbol{\theta}}^{(t)}(\boldsymbol{x}_t)}{\sqrt{1-\bar{\alpha}_t}}$ (Song & Ermon, 2019). Once the score function is

estimated by $s_\theta$ for all $t$, we can compute Equation 4 and simulate the reverse SDE to reconstruct the data sample from $p_0$. Despite its high quality, it is notorious for its slow sampling process. To address this, (Song et al., 2021a) proposes denoising diffusion implicit models (DDIM) that accelerate the sampling process based on non-Markovian assumptions. When $\bar{\alpha}_t = \prod_{i=1}^{t}(1 - \beta_i)$, the sampling process is given by

$$\boldsymbol{x}_{t-1} = \sqrt{\bar{\alpha}_{t-1}}\hat{\boldsymbol{x}}_t + \sqrt{1 - \bar{\alpha}_{t-1}}\hat{\boldsymbol{\epsilon}}_t \tag{6}$$

where $\hat{\boldsymbol{x}}_t = \frac{\boldsymbol{x}_t - \sqrt{1-\bar{\alpha}_t}\boldsymbol{\epsilon}_{\boldsymbol{\theta}}^{(t)}(\boldsymbol{x}_t)}{\sqrt{\bar{\alpha}_t}}$ is the denoised estimate of $\boldsymbol{x}_t$ derived from Tweedie's formula (Efron, 2011), and $\hat{\boldsymbol{\epsilon}}_t = \frac{\sqrt{1-\bar{\alpha}_{t-1}-\sigma_t^2}\boldsymbol{\epsilon}_{\boldsymbol{\theta}}^{(t)}(\boldsymbol{x}_t) + \sigma_t\boldsymbol{\epsilon}}{\sqrt{1-\bar{\alpha}_{t-1}}}$ represents the noise term at time step $t$, which is a weighted combination of the deterministic $\boldsymbol{\epsilon}_{\boldsymbol{\theta}}^{(t)}(\boldsymbol{x}_t)$ and stochastic $\boldsymbol{\epsilon} \sim \mathcal{N}(\boldsymbol{0}, \boldsymbol{I})$ component. The parameter $\sigma_t$ affects the sampling process and is often set to be $\sigma_t = \eta\sqrt{\frac{1-\bar{\alpha}_{t-1}}{1-\bar{\alpha}_t}}\sqrt{1 - \frac{\bar{\alpha}_t}{\bar{\alpha}_{t-1}}}$. Especially, when $\eta = 0$, the sampling becomes fully deterministic, while $\eta = 1$ results in a sampling process equivalent to denoising diffusion probabilistic models (DDPM) (Song et al., 2021a).

On the other hand, to solve inverse problems, we need to sample the solution $\boldsymbol{x}$ from the posterior distribution $\boldsymbol{x} \sim p(\boldsymbol{x}|\boldsymbol{y})$. With diffusion models, the score function $\nabla_{\boldsymbol{x}_t} \log p_t(\boldsymbol{x}_t)$ in Equation 4 should be replaced by posterior score $\nabla_{\boldsymbol{x}_t} \log p_t(\boldsymbol{x}_t|\boldsymbol{y})$ (Chung et al., 2022; 2023; Song et al., 2023),

$$d\boldsymbol{x}_t = \Big[ - \frac{\beta_t}{2}\boldsymbol{x_t} - \beta_t\nabla_{\boldsymbol{x}_t} \log p_t(\boldsymbol{x}_t|\boldsymbol{y}) \Big]dt + \sqrt{\beta_t}d\bar{\boldsymbol{w}}. \tag{7}$$

From Bayes' rule, the posterior can be decomposed as $p(\boldsymbol{x}|\boldsymbol{y}) \propto p(\boldsymbol{x})p(\boldsymbol{y}|\boldsymbol{x})$, and $\nabla_{\boldsymbol{x}_t} \log p_t(\boldsymbol{x}_t)$ is readily replaced by the posterior

$$\nabla_{\boldsymbol{x}_t} \log p_t(\boldsymbol{x}_t|\boldsymbol{y}) = \nabla_{\boldsymbol{x_t}} \log p_t(\boldsymbol{x}_t) + \nabla_{\boldsymbol{x}_t} \log p_t(\boldsymbol{y}|\boldsymbol{x}_t), \tag{8}$$

where it is required to compute both the prior term $\nabla_{\boldsymbol{x}_t} \log p_t(\boldsymbol{x}_t)$, and the likelihood term $\nabla_{\boldsymbol{x}_t} \log p_t(\boldsymbol{y}|\boldsymbol{x}_t)$. While the score function $\nabla_{\boldsymbol{x}_t} \log p_t(\boldsymbol{x}_t)$ can be obtained from pre-trained score networks $s_\theta(\boldsymbol{x}_t, t)$, the likelihood term $\nabla_{\boldsymbol{x}_t} \log p_t(\boldsymbol{y}|\boldsymbol{x}_t)$ is usually intractable. To address this issue, (Chung et al., 2023) proposed diffusion posterior sampling (DPS) that uses a Gaussian approximation for the likelihood term, which results in a one-step gradient update

$$\nabla_{\boldsymbol{x}_t} \log p_t(\boldsymbol{y}|\boldsymbol{x}_t) \approx \rho\nabla_{\boldsymbol{x}_t}\big\|\boldsymbol{y} - \mathcal{A}(\hat{\boldsymbol{x}}_0^{(t)})\big\|_2^2, \tag{9}$$

where $\rho$ is the step size controlling data-consistency strength and $\hat{\boldsymbol{x}}_0^{(t)}$ is denoised estimate. Although theoretically sound, updating the likelihood at every reverse sampling time step is inefficient due to the tradeoff between computational cost and accuracy, especially when the forward model $\mathcal{A}$ is expensive and highly nonlinear. Therefore, we introduce a physics-constrained diffusion model (PCDM) to address this issue by treating the prior and likelihood terms separately enhancing the speed by using an accelerated diffusion sampler, and improving the accuracy by solving the likelihood objective by allowing multiple iterations.

## 3.2 VARIABLE SPLITTING

To separate the likelihood term and prior term in Equation 2, we draw inspiration from the classic variable splitting method, such as half quadratic splitting (HQS) and alternating direction method of multipliers (ADMM) (Boyd et al., 2011; Venkatakrishnan et al., 2013; Zhang et al., 2017; 2021; Li et al., 2024). This approach introduces an auxiliary variable $z$ in Equation 2,

$$\min_{\boldsymbol{x}} \frac{1}{2}\|\boldsymbol{y} - \mathcal{A}(\boldsymbol{x})\|_2^2 + \lambda\mathcal{R}(\boldsymbol{z}), \qquad s.t. \quad \boldsymbol{z} = \boldsymbol{x}. \tag{10}$$

From HQS, the objective function to optimize can be reformulated as

$$\mathcal{L}_\mu(\boldsymbol{z}, \boldsymbol{x}) = \frac{1}{2}\|\boldsymbol{y} - \mathcal{A}(\boldsymbol{x})\|_2^2 + \lambda\mathcal{R}(\boldsymbol{z}) + \frac{\mu}{2}\|\boldsymbol{z} - \boldsymbol{x}\|_2^2, \tag{11}$$

where $\mu$ is a penalty coefficient. Equation 11 can be solved by alternating optimizing the following subproblems for $\boldsymbol{z}$ and $\boldsymbol{x}$, while keeping other variable fixed:

$$\boldsymbol{z}_{i+1} = \arg\min_{\boldsymbol{z}}\mathcal{L}_\mu(\boldsymbol{z}, \boldsymbol{x}_i), \tag{12}$$

$$\boldsymbol{x}_{i+1} = \arg\min_{\boldsymbol{x}}\mathcal{L}_\mu(\boldsymbol{z}_{i+1}, \boldsymbol{x}) \tag{13}$$

This technique decouples the likelihood terms $\|\boldsymbol{y} - \mathcal{A}(\boldsymbol{x})\|_2^2$ and the prior terms $\mathcal{R}(\boldsymbol{x})$, separately. In our scenarios, the prior terms, treated by Equation 12, make the solution meaningful following the data distribution learned from diffusion models, Meanwhile, the likelihood terms, optimized via Equation 13, ensure that the solution strictly adheres to the given physics constraints.

### 3.3 REGUARLIZING VIA DIFFUSION SAMPLING

Instead of using a traditional regularizer, we can employ pre-trained diffusion models as an implicit regularizer. We utilize an accelerated sampler, such as DDIM, to avoid slow sampling, as discussed in the preliminary section. We set the time steps $0 = t_0 < \cdots < t_k < t_{k+1} < \cdots < t_K = T$ as a subset of $[0, T]$, with $\boldsymbol{z}_T$ and $\boldsymbol{x}_T$ initialized from the a Gaussian distribution. For readability, we denote $x_{t_k}$ as $x_k$ for readability. Equation 12 can then be replaced by the following two-step sampling rule:

$$\boldsymbol{z}_k' = \sqrt{\bar{\alpha}_k}\boldsymbol{x}_{k+1} + \sqrt{1 - \bar{\alpha}_k}\hat{\boldsymbol{\epsilon}}_{k+1}, \qquad \boldsymbol{z}_k = \frac{\boldsymbol{z}_k' - \sqrt{1 - \bar{\alpha}_k}\boldsymbol{\epsilon}_{\boldsymbol{\theta}}^{(k)}(\boldsymbol{z}_k')}{\sqrt{\bar{\alpha}_{t_k}}}. \tag{14}$$

We first sample from the previous optimized denoised estimate $\boldsymbol{x}_{k+1}$ to obtain the noisy data at the next time step $t_k$, resulting in $\boldsymbol{z}_k'$. The purpose of this procedure is to transition the sample from the denoised manifold (time step $t_0 = 0$) to the manifold of the noise level at the subsequent time step $t_k$. Next, we compute the denoised estimate of $\boldsymbol{z}_k'$ using Tweedie's formula (Efron, 2011), yielding $\boldsymbol{z}_k$. This procedure aims to transition the sample from a manifold of the noise level at time step $t_k$ back to a denoised manifold (time step $t_0 = 0$) for further optimization in physical space. While reducing the number of reverse steps with accelerated diffusion samplers can speed up the inference time, it may cause convergence issues when the effective number of likelihood updates is insufficient to fully optimize the objective. Therefore, it is necessary to strictly enforce physics constraints after each prior step.

### 3.4 ENFORCING PHYSICS CONSTRAINTS

From Equation 13, we derive the following optimization problem:

$$\boldsymbol{x}_k = \arg\min_{\boldsymbol{x}}\|\boldsymbol{y} - \mathcal{A}(\boldsymbol{x})\|_2^2 + \mu\|\boldsymbol{z}_k - \boldsymbol{x}\|_2^2 \tag{15}$$

which aims to find a proximal solution of $z_k$ while ensuring that the solution strictly adheres to the physics constraints. In our scenarios, due to the complexity and highly nonlinear nature of the physics constraints $\mathcal{A}$, obtaining a closed-form solution for Equation 15 is usually not available (Vono et al., 2022). Instead, we solve the optimization problem using the denoised estimates $\boldsymbol{z}_k$ from the prior steps as an initial guess

$$\boldsymbol{x}_k^{(n+1)} = \boldsymbol{x}_k^{(n)} - \alpha\nabla_{\boldsymbol{x}_k^{(n)}}\|\boldsymbol{y} - \mathcal{A}(\boldsymbol{x}_k^{(n)})\|_2^2, \qquad \boldsymbol{x}_k^{(0)} = \boldsymbol{z}_k, \tag{16}$$

where $\alpha$ is the step size of the likelihood updates, and $N$ is selected to the certain number of the inner updates $0 \leq n \leq N$ within a single likelihood step. Instead of using coefficient $\mu$ to balance the measurement-consistency term and proximal term, our likelihood updates start from the previous denoised estimate $\boldsymbol{z}_k$ and set the number of likelihood (inner) updates for searching the solution near the $\boldsymbol{z}_k$. Additionally, unlike a one-step likelihood gradient update, our iterative approach to solving the optimization problem enhances convergence and avoids the computation for coupling and enforces physical constraints more strictly through $N$ inner iterations starting from the previous steps. This approach increases stability and makes it well-suited for large-scale inverse problems (Chung et al., 2024; Song et al., 2024; Li et al., 2024). In our experiments, we utilize the Adam optimizer (Kingma, 2014), and the optimization process is conducted after a certain time step $t \leq t_s$ since the enforcement has only marginal effects during the early stage of the diffusion reverse steps (Yu et al., 2023; Song et al., 2024). For example, when $t_s = 0.5T$, the likelihood update processes are conducted only at $t < 0.5T$ of reverse steps, while only the unconditional reverse steps are conducted for $t > 0.5T$. Instead of tuning hyperparameters like $\mu$ and $\lambda$, the step size for inner updates $\alpha$, the number of inner updates $N$, and the starting time step $t_s$ are employed as the hyperparameters.

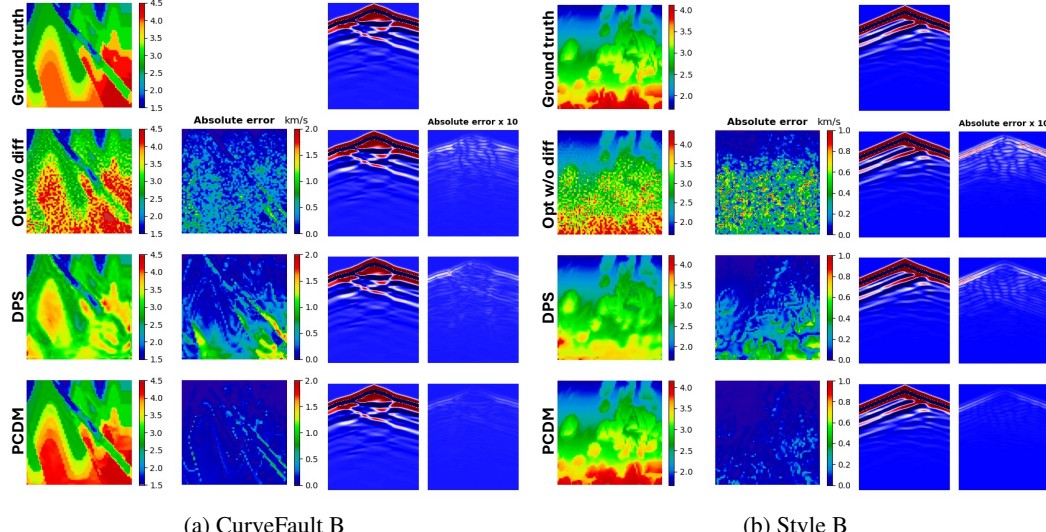

(a) CurveFault B                    (b) Style B

Figure 2: Qualitative results for full-waveform inversion: (a) CurveFault B and (b) Style B. In each figure, the first and second columns display the ground truth and predicted velocity fields $\boldsymbol{x}$, along with the corresponding absolute error. The third and fourth columns show the given measurements $y$, the estimated measurements $\mathcal{A}(\boldsymbol{x})$, and the absolute error of the residual $\|\boldsymbol{y}-\mathcal{A}(\boldsymbol{x})\|$, respectively.

| Method | CurveFault B | | | | Style B | | | |
|---|---|---|---|---|---|---|---|---|
| | Res | MAE ↓ | RMSE ↓ | SSIM ↑ | Res | MAE ↓ | RMSE ↓ | SSIM ↑ |
| InversionNet | - | 1.67e-1 | 2.41e-1 | 0.605 | - | 5.86e-2 | 8.93e-2 | 0.760 |
| VelocityGAN | - | 1.58e-1 | 2.34e-1 | 0.603 | - | 6.49e-2 | 9.79e-2 | 0.725 |
| Opt w/o diff | 1.42e-3 | 2.17e-1 | 3.16e-1 | 0.410 | 2.53e-4 | 2.36e-1 | 3.25e-1 | 0.294 |
| DPS (1000) | 6.79e-4 | 1.29e-1 | 2.38e-1 | 0.632 | 1.49e-4 | 9.14e-2 | 1.35e-1 | 0.593 |
| PCDM (200) | **3.57e-5** | **4.89e-2** | **9.91e-2** | **0.850** | **4.96e-5** | **3.07e-2** | **5.37e-2** | **0.890** |

Table 1: Quantitative results for FWI. Numbers in parentheses represent the number of reverse steps. Roughly speaking, DPS (1000) involves 1000 likelihood steps, PCDM (200) includes 1000 likelihood iterations (likelihood steps starting after $t_s = 200/2$ and at most 10 iterations per reverse step), and Opt w/o diff also involves 1000 likelihood iterations (same number of PCDM).

## 4 EXPERIMENTS

### 4.1 FULL-WAVEFORM INVERSION

Full-waveform inversion (FWI) is an example of the physics-constrained inversion problem that aims to obtain geophysical properties ($\boldsymbol{x} = v(\boldsymbol{r})$) from seismic measurements ($\boldsymbol{y} = p(\boldsymbol{r},t)$) which is governed by the acoustic wave equation ($\mathcal{A}$),

$$\left(\nabla^2 - \frac{1}{v(\boldsymbol{r})^2}\frac{\partial^2}{\partial t^2}\right)p(\boldsymbol{r},t) = s(\boldsymbol{r},t), \tag{17}$$

where $\nabla^2 = \frac{\partial^2}{\partial x^2} + \frac{\partial^2}{\partial y^2} + \frac{\partial^2}{\partial z^2}$ is laplace operator, $p(\boldsymbol{r},t)$ represents the pressure wavefield at spatial position $\boldsymbol{r}$ and time $t$, $v(\boldsymbol{r})$ denotes velocity field, and $s(\boldsymbol{r},t)$ is the source function. We utilize the large-scale seismic benchmark dataset, OpenFWI, which consists of given measurements and their corresponding velocity fields, as detailed in (Deng et al., 2022). For the forward model, we utilized the open-sourced Deepwave package (Richardson, 2023), which implements the discretized wave propagation using PyTorch (Paszke et al., 2019). The dataset consists of various families of velocity fields, each characterized by distinct structural features and velocity variations with depth. Each family has two versions: an easier version (A) and a more difficult version (B). While existing frameworks generally work well on version A, they often struggle with version B (Deng et al., 2022). To highlight the superiority of our methods, we choose version B of two velocity field families,

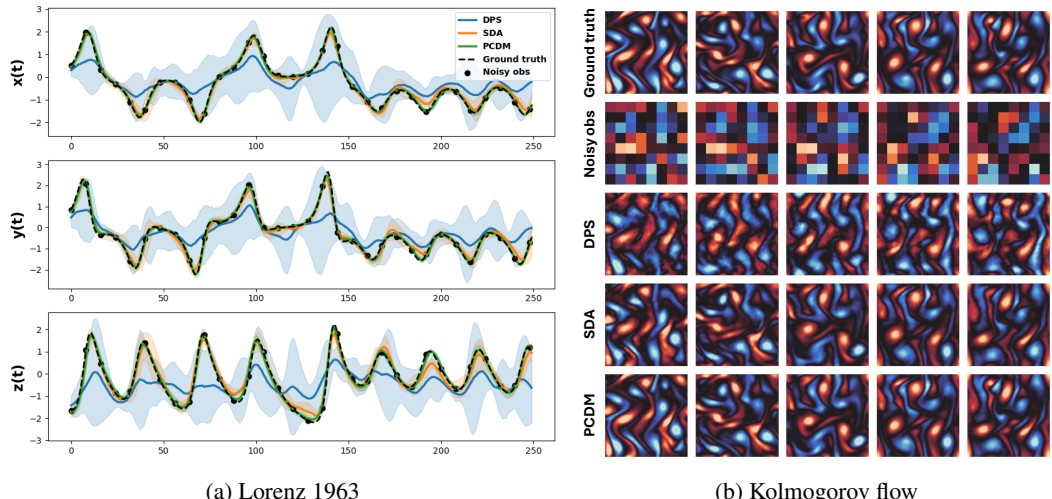

(a) Lorenz 1963          (b) Kolmogorov flow

Figure 3: Qualitative results for data assimilation (DA): (a) Lorenz 1963 and (b) Kolmogorov flow. In (a), the black dotted line and scattered points are the ground truth and noisy observations. Colored lines and shaded uncertainty bands show the mean and standard deviation computed from 5 generated samples from each method. In (b), the first and second rows depict the ground truth and noisy observations of the vorticity fields, $\boldsymbol{\omega} = \nabla \times \boldsymbol{u}$. The subsequent rows present the predictions for each method based on the given observations.

| Method | Lorenz 1963 | | | Method | Kolmogorov flow | | |
|---|---|---|---|---|---|---|---|
| | MAE ↓ | RMSE ↓ | Time(s) | | MAE ↓ | RMSE ↓ | Time(s) |
| DPS (500) | 9.47e-1 | 1.20e0 | 9.4 | DPS (1000) | 2.22e-1 | 3.98e-1 | 45.7 |
| SDA (500) | 4.49e-2 | 5.86e-2 | 9.7 | SDA (1000) | 8.18e-2 | 1.12e-1 | 45.6 |
| PCDM (100) | **1.49e-2** | **2.22e-2** | **2.4** | PCDM (200) | **2.07e-2** | **2.70e-2** | **11.2** |

Table 2: Quantiative results for DA. Numbers in parentheses represent the number of reverse steps.

"CurveFault B" and "Style B". We take InversionNet (Wu & Lin, 2020) and VelocityGAN (Zhang & Lin, 2020) as baselines of supervised end-to-end frameworks that directly learn the mapping between input and output without incorporating physics constraints. Additionally, we use DPS (Chung et al., 2023) as an unsupervised plug-and-play baseline, which leverages a learned generative prior but relies on one-step coupled likelihood gradient updates at each reverse timestep. For further comparison, we include optimization without diffusion prior (opt w/o diff), which performs the same number of optimizing steps as our PDCM but without utilizing diffusion priors. We use the same train/test split and evaluation metrics between ground truth velocity and predicted velocity, including mean absolute error (MAE), rooted mean squared error (RMSE), and structural similarity (SSIM), as described in the (Deng et al., 2022). Additionally, we present the residual $\|\boldsymbol{y} - \mathcal{A}(\boldsymbol{x})\|$, which presents the difference between the true seismic measurements $\boldsymbol{y}$ and the estimated measurements from forward physics and predicted velocity $\mathcal{A}(\boldsymbol{x})$.

As shown in Table 5, PCDM outperforms all comparisons across all metrics. Figure 7 illustrates that optimization without diffusion (Opt w/o diff) struggles with local minima due to a lack of appropriate regularization. DPS is suboptimal because the effective number of likelihood steps is insufficient to fully optimize the physics-constrained inverse problem. In contrast, PCDM provides plausible solutions by effectively integrating diffusion models and physics constraints, resulting in efficiently navigating toward the global minimum for the inverse problems.

## 4.2 DATA ASSIMILATION

Data assimilation (DA) can be considered as a physics-constrained inverse problem that aims to estimate the states of a system $(\boldsymbol{x})$ by integrating partial observational data $(\boldsymbol{y})$ with underlying physical models $(\mathcal{A} = \{\mathcal{M}, \mathcal{P}\})$, which serve as constraints. In our scenario, we have two types of

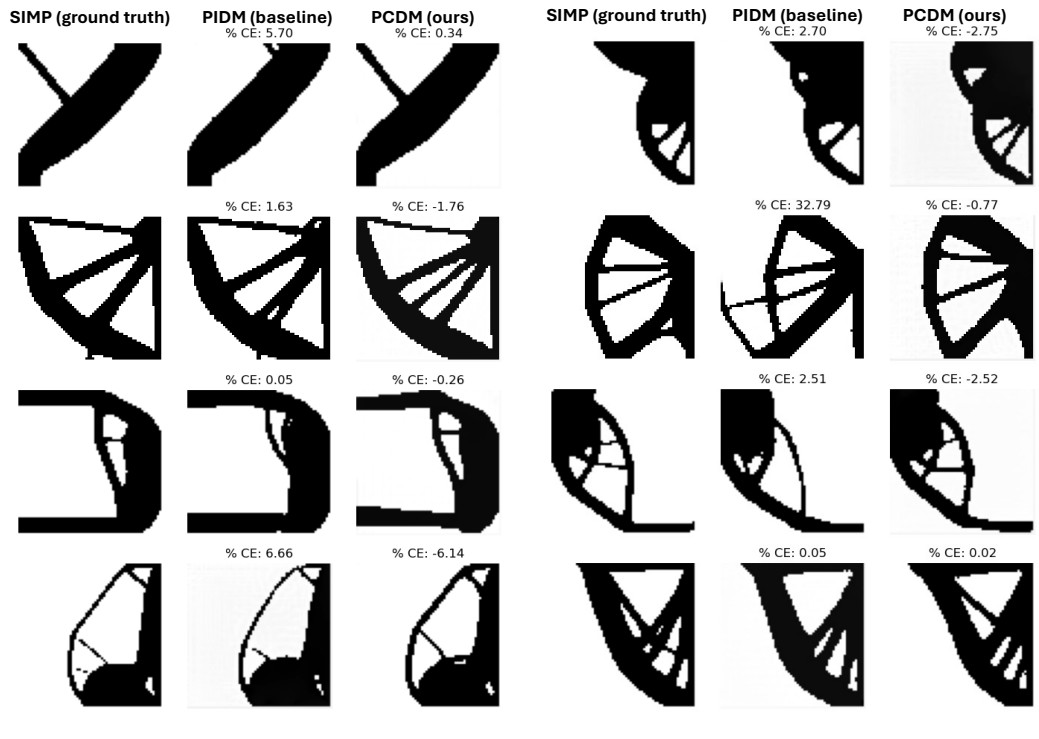

(a) In-distribution                          (b) Out-of-distirbution

Figure 4: Qualitative results for topology optimization (TO): testsets for (a) in- and (b) out-of-distribution. % CE indicates the compliance error relative to the ground truth. Note that the negative value implies that the generated solutions have lower compliance than those produced by SIMP.

| Method | In-distribution | | | Out-of-distribution | | |
| --- | --- | --- | --- | --- | --- | --- |
| | Res ↓ | MDN % CE ↓ | % VFE ↓ | Res ↓ | MDN % CE ↓ | % VFE ↓ |
| TopoDiff-G | - | 0.83 | **1.49** | - | 1.82 | 1.80 |
| DOM | - | 0.74 | 1.52 | - | 3.47 | **1.59** |
| PIDM | 1.24e-3 | 0.06 | 2.25 | 1.29e-3 | 0.56 | 1.91 |
| PCDM | **6.85e-4** | **-0.82** | 1.50 | **7.61e-4** | **0.05** | 1.79 |

Table 3: Quantitative results for TO. The terms Res, MDN % CE, and % VFE represent the median residual error, median compliance error, and mean volume fraction error, respectively.

constraints: the first is observational data from sparse measurements $\boldsymbol{y} = \mathcal{M}(\boldsymbol{x})$, and the second is the physical residual, $r = \mathcal{P}(\boldsymbol{x}) = 0$. The datasets we considered are two time-dependent physical models ($\mathcal{P}$), Lorenz 1963, and Kolmogorov flow. We follow the details provided in (Rozet & Louppe, 2023) for the benchmarks. The governing equations for the Lorenz 1963 are given by

$$\frac{dx}{dt} = \sigma(y - x), \quad \frac{dy}{dt} = x(\rho - z) - y, \quad \frac{dz}{dt} = xy - \beta z, \quad (18)$$

where the system parameters $\sigma = 10$, $\rho = 28$, and $\beta = \frac{8}{3}$ result in chaotic behavior. To compute the residuals $r = \mathcal{P}(\boldsymbol{x})$, we take the 2nd-order central finite difference to calculate the time derivative of $\frac{d}{dt}[x, y, z]$. Following the dataset generation as detailed in (Rozet & Louppe, 2023), we use 1024 independent trajectories, each of length 1024, generated from various initial states. We maintain the same train/test split described in (Rozet & Louppe, 2023). For the Lorenz 1963 scenario, the partial observations $\boldsymbol{y}$ are given by 8x coarsening the original data with noise added with $\sigma = 0.25$.

On the other hand, Kolmogorov flow represents an incompressible fluid governed by the Navier-Stokes equations:

$$\frac{d\boldsymbol{u}}{dt} = -\boldsymbol{u}\nabla\boldsymbol{u} + \frac{1}{Re}\nabla^2\boldsymbol{u} - \frac{1}{\rho}\nabla p + \boldsymbol{f}, \quad \nabla \cdot \boldsymbol{u} = 0. \quad (19)$$

where $\boldsymbol{u}$ is the velocity field, $Re$ is the Reynold number, $\rho$ is the fluid density, $p$ is the pressure field and $f$ represents external forcing. To compute the residuals $r = \mathcal{P}(\boldsymbol{x})$, the time derivative is obtained using three consecutive frames. The convection and diffusion terms in Equation 19 are calculated by approximating the Laplacian and gradient of vorticity in Fourier space, followed by transforming them back to the physical space, as described in (Shu et al., 2023). Following the dataset generation as detailed in (Rozet & Louppe, 2023), we utilize 1024 independent trajectories, each of length 64, where each state is represented as 64x64x2 with two velocity channels, generated from randomly sampled initial states. In the case of Kolmogorov flow, the partial observations $\boldsymbol{y}$ are obtained by 8x spatially coarsening and 4x temporal coarsening to the original data with noise added with $\sigma = 0.1$. Consequently, each partial observation has a state represented as 8x8x2. We use DPS (Chung et al., 2023) and SDA (Huang et al., 2024) as our baselines. SDA is a variant of DPS that rescales the likelihood score to stabilize the sampling process. Both methods leverage a learned generative prior and rely on one-step likelihood updates at each reverse process. In contrast, PCDM allows multi-step likelihood updates at each reverse timestep, maintaining effectiveness even with fewer timesteps, such as using DDIM.

As demonstrated in Table 2, PCDM outperforms all comparisons across all metrics including MAE, RMSE, and inference time. In particular, PCDM achieves superior accuracy on benchmarks with 5 times fewer reverse steps, indicating that it provides more physically plausible solutions with faster inference. In contrast, predictions from other methods either deviate significantly from the ground truth, as shown in Figure 3 (a) or lack physical consistency, as illustrated in Figure 3 (b).

### 4.3 TOPOLOGY OPTIMIZATION

Topology optimization (TO) is another example of the physics-constrained inverse design that aims to identify an optimal physical structure $(\boldsymbol{x})$ that satisfies elastic equilibrium $(\mathcal{A} = \{\mathcal{C}, KU = F\})$, with given loads and boundary conditions $(\boldsymbol{y} = \{F, V_0\})$. The problem can be represented as

$$\min_{\boldsymbol{x}} \mathcal{C}(\boldsymbol{x}) = F^T U(\boldsymbol{x}), \qquad s.t. \quad K(\boldsymbol{x})U(\boldsymbol{x}) = F, \quad V(\boldsymbol{x}) \leq V_0, \quad 0 \leq x_{ij} \leq 1. \quad (20)$$

where $\mathcal{C}$ is compliance as the objective, $F$ is applied loads, $U(\boldsymbol{x})$ is the node displacement, $K(\boldsymbol{x})$ is the stiffness matrix, $V(\boldsymbol{x})$ and $V_0$ are the volume fraction and volume constraint, and the design variables $x_{ij}$ are continuous value between 0 and 1. This problem is traditionally solved using the Solid Isotropic Material with Penalization (SIMP) method, which is based on the finite element method (FEM) (Bendsoe & Sigmund, 2013). We utilize a dataset composed of given constraints and their corresponding optimal topologies solved by the SIMP as our ground truth described in (Mazé & Ahmed, 2023). The dataset includes 30,000 optimal topologies with various boundary conditions and two levels of test sets with in-distribution and out-of-distribution boundary conditions. We compare PCDM with state-of-the-art approaches that employ diffusion models for topology optimization. TopoDiff-G (Mazé & Ahmed, 2023) introduces a diffusion model guided by a gradient update from auxiliary surrogate models in every, which helps to reduce compliance and enforces boundary conditions at each step of the reverse sampling process. DOM (Giannone et al., 2023) aligns the denoising trajectory with the optimization trajectory of the traditional iterative solver. PIDM (Bastek et al., 2024) proposes a novel physics-informed framework during the training phase, but not applying correction during inference. We follow the same train/test split and evaluation metrics as described in (Mazé & Ahmed, 2023; Giannone et al., 2023; Bastek et al., 2024) to ensure fair comparisons. The evaluation metrics include the median residual error of the predicted solution (Res), which quantifies the extent of violating of elastic equilibrium equation, the median compliance error (MDN % CE) which is relative to the ground truth, $\mathrm{CE} = (\mathcal{C}(\boldsymbol{x}) - \mathcal{C}(\boldsymbol{x}^*))/\mathcal{C}(\boldsymbol{x}^*)$, and the mean volume fraction error (% VFE), which is relative to the input volume fraction, $\mathrm{VFE} = |V(\boldsymbol{x}) - V(\boldsymbol{x}^*)|/V(\boldsymbol{x}^*)$.

As shown in Table 3, PCDM significantly outperforms existing methods in residual and compliance errors, demonstrating that our approach ensures elastic equilibrium and produces more stable structures. Notably, our method achieves a negative compliance error across in-distribution test sets and examples illustrated in Figure 4. This indicates that the generated topology exhibits lower compliance, i.e. a more stable structure under the given boundary conditions compared to the topology produced by SIMP. By strictly enforcing physics constraints and utilizing efficient diffusion sampling, our method consistently provides superior structures compared to existing methods.

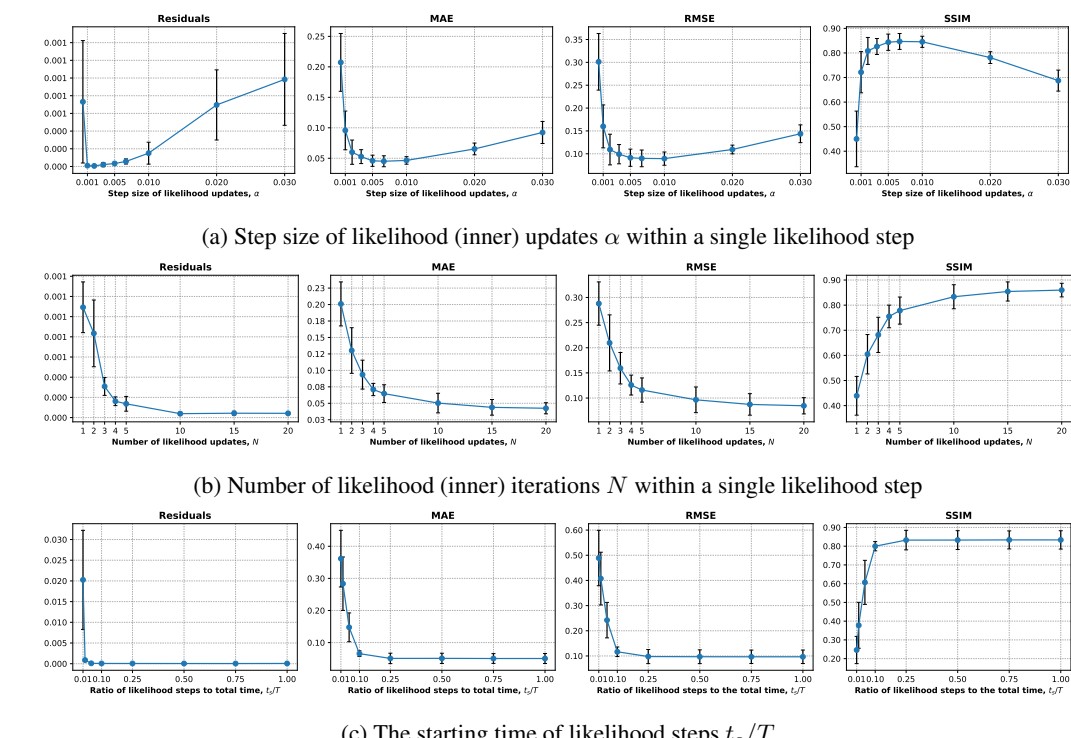

(a) Step size of likelihood (inner) updates $\alpha$ within a single likelihood step

(b) Number of likelihood (inner) iterations $N$ within a single likelihood step

(c) The starting time of likelihood steps $t_s/T$

Figure 5: Ablation studies on (a) step size $\alpha$, (b) the number of likelihood (inner) updates $N$ within a single likelihood step, and (c) the starting time of likelihood steps $t_s/T$.

### 4.4 ABLATION STUDIES

In Figure 5, we conduct ablation studies on the step size $\alpha$, the number of likelihood (inner) iterations $N$ within a single likelihood step, and the starting time of likelihood steps $t_s/T$ on the CurveFaultB benchmark for the full waveform inversion problem. The results demonstrate that selecting an appropriate step size, such as $\alpha = 5e - 3$ in this case, is essential for obtaining optimal solutions; otherwise, the performance decreases. Performing more than 10 likelihood iterations per likelihood step achieves sufficient solution accuracy. While higher update counts offer only slight improvements at the cost of increased computational time. Additionally, starting likelihood steps after $t_s/T = 25\%$ of the reverse process achieves stable convergence. This demonstrates that conducting likelihood steps during only the final $25\%$ of reverse steps is sufficient, significantly reducing computational times. This reduction is due to the marginal impact of likelihood steps during the early stages of diffusion reverse steps, as discussed in (Yu et al., 2023; Song et al., 2024).

## 5 CONCLUSION

In this paper, we presented the physics-constrained diffusion model (PCDM), a framework designed to solve the inverse problems in scientific and engineering domains effectively. By integrating pretrained diffusion models with physics-constrained objectives, PCDM provides plausible and physically consistent solutions in a feasible inference time. Our approach leverages accelerated diffusion sampling, enabling effective reverse steps in fewer timesteps, while strictly adhering to physics constrained by multi-step optimizing the likelihood objective at each reverse timestep. Extensive experiments on a variety of challenging physics-constrained inverse problems, including full-waveform inversion, data assimilation, and topology optimization, demonstrate that our method consistently outperforms existing approaches. The results highlight PCDM's ability to provide high-quality solutions without compromising computational complexity, making it a practical tool for real-world applications in fields where complex and nonlinear physics are prevalent.

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

## A IMPLEMENTATION DETAILS

### A.1 PROBLEM DETAILS

**Full waveform inversion.** We used CurveFault B and Style B datasets from the OpenFWI benchmarks (Deng et al., 2022). The original velocity field represents a 700 m×700 m, discretized on a 70×70 grid with 10 m grid spacings. We cropped 3 pixels from each boundary, reducing the grid size to 64×64 and the corresponding velocity field 640 m×640 m which represent $v(r)$ in equation 17. We set the 5 sources located from $x = 0$ m to $x = 640$ m with evenly spacing $dx = 160$ m at a depth of $z = 10$ m, and the 64 receivers located from $x = 0$ to $x = 640$ m with spacing $dx = 10$ m at the same depth of $z = 10$ m. The Ricker wavelet with a central frequency of 15 Hz is used as the source function $s(r,t)$ in equation 17. Simulations were conducted for 1 s with a time step of 0.001 s, resulting in measurements $y$ with the size of 5×64×1000. In this case, we make 5 batches of gradient updates for 1 likelihood iterations. We utilize the open-sourced Deepwave package (Richardson, 2023), which implements the discretized wave equation using PyTorch (Paszke et al., 2019) via the finite difference method. A 4th-order finite difference scheme was employed, along with a perfectly matched layer (PML) of 120 grid points to prevent reflections from the edges of the model. Due to the significant discrepancy in value ranges - where the true velocity fields $v(r)$ typically span from 1500 m/s to 4500 m/s, while random initialization of the diffusion models or optimization solvers usually follows a standard normal distribution $\mathcal{N}(0,1)$ - we preprocess the training set of velocity fields with min-max normalization before pre-training the diffusion model. During the inference stage, at each likelihood step, the output of the previous diffusion reverse step is denormalized to compute the measurement-consistency term. Subsequently, the updated state of the likelihood step is normalized back into the appropriate range of values for the diffusion models. For the measurement-consistency term, we employed a combination of $l_1$ and $l_2$ loss function (Deng et al., 2022). Specifically, the term is expressed as $c_1 \cdot \|\boldsymbol{y} - \mathcal{A}(\boldsymbol{x})\|_1 + c_2 \cdot \|\boldsymbol{y} - \mathcal{A}(\boldsymbol{x})\|_2$, with $c_1$ and $c_2$ both set to 0.5.

**Data assimilation.** We used the Lorenz 1963 and Kolmogorov flow datasets as benchmarks, following (Rozet & Louppe, 2023). For Lorenz 1963, the system parameters were set to $\sigma = 10$, $\rho = 28$, and $\beta = \frac{8}{3}$. The partial observations $y$ are given by 8x coarsening the original data with noise added with $\sigma_n = 0.25$. The time derivative of the state for residual computations was computed by the 2nd-order central finite difference. For the Kolmogorov flow, the state consists of 2-dimensional velocity fields of size 64×64×2 within the domain $[0, 2\pi]^2$ with periodic boundary conditions. The Reynolds number was set to $Re = 1000$, the fluid density $\rho = 1$, and external forcing $f$ followed Kolmogorov forcing with linear damping (Chandler & Kerswell, 2013). The partial observations $y$ are obtained by 8x spatially coarsening and 4x temporal coarsening to the original data with noise added with $\sigma_n = 0.1$. To compute the residuals, the time derivative is obtained using three consecutive frames. The convection and diffusion terms in equation 19 are calculated by approximating the Laplacian and gradient of vorticity in Fourier space, followed by transforming them back to the physical space, described in (Shu et al., 2023). Data generation and implementation for both datasets were based on (Rozet & Louppe, 2023).

**Topology optimization.** We used topology optimization benchmarks (Mazé & Ahmed, 2023) which include 30,000 topologies for training, about 1,800 new topologies for the in-distribution test set, and 1,000 new topologies for the out-of-distribution test set. In this problem, it includes three constraints compliance $\mathcal{C}(x) = U^T(x)K^T(x)U(x)$, elastic equilibrium $K(x)U(x) = F$, and volume constraint, $\frac{1}{N}\sum_i x_i \leq V_0$, where $K(x)$ and $U(x)$ are the global stiffness and displacement respectively, and $F$ is given loads. We implement the physical constraint terms as, $\|K(x)U(x) - F\|_2^2 + c_1 \cdot \|\mathcal{C}(x) - 0\|_2^2 + c_2 \cdot ReLU(\frac{1}{N}\sum_i x_i - V_0)$, where given loads and volume conditions can be considered as boundary conditions $y = \{0, F, V_0\}$ and the compliance and elastic equations can be considered as forward operator $\mathcal{A} = \{\mathcal{C}, KU, \frac{1}{N}\sum_i x_i\}$. We set the coefficients with $c_1 = 1e-4$ and $c_2 = 1$.

### A.2 ARCHITECTURE AND TRAINING

The implemented details of architectures and training for each problem are summarized in Table 4 respectively. We adopt the architecture and training details for the data assimilation and topology optimization from (Rozet & Louppe, 2023) and (Bastek et al., 2024), respectively.

Table 4: Details of architectures and training.

| Problem | Full waveform inversion | | Data assimilation | | Topology optimization | |
|---|---|---|---|---|---|---|
| Dataset | CurveFault B | Style B | Lorenz 1963 | Kolmogorov flow | In-distribution | Out-of-distribution |
| Architecture | | | | | | |
| Model | U-Net | U-Net | MLP | U-Net | U-Net | U-Net |
| Target resolutions | $64 \times 64$ | $64 \times 64$ | 3 | $64 \times 64 \times 2$ | $64 \times 64$ | $64 \times 64 \times 3$ |
| Latent channels | 128 | 128 | 64 | [32, 64, 128] | [128, 256, 512] | [128, 256, 512] |
| Attention resolution | 16 | 16 | - | - | 32 | 32 |
| Number of residual blocks | 2 | 2 | 3 | 3 | 2 | 2 |
| Activation | SiLU | SiLU | SiLU | SiLU | SiLU | SiLU |
| Normalization | LayerNorm | LayerNorm | LayerNorm | LayerNorm | LayerNorm | LayerNorm |
| Training | | | | | | |
| Optimizer | Adam | Adam | Adam | Adam | Adam | Adam |
| Batch size | 128 | 128 | 64 | 64 | 8 | 8 |
| Timesteps | 1000 | 1000 | 500 | 1000 | 1000 | 1000 |
| $\beta$ schedule | Linear | Linear | Linear | Linear | Cosine | Cosine |
| Epochs | 1000 | 1000 | 500 | 1000 | 1000 | 1000 |
| Learning rate | 1e-4 | 1e-4 | 1e-3 | 2e-4 | 1e-4 | 1e-4 |
| Weight decay | 0.5 | 0.5 | 1e-3 | 1e-3 | - | - |

## A.3 ALGORITHMS

**Opt w/o diff** We employ the Adam optimizer (Kingma, 2014) with a learning rate of 0.005 and perform 1,000 iterations for the total optimization process. For the initialization, we take a random initialization from the standard normal distribution $\mathcal{N}(0, 1)$. During the optimization, we denormalize the $x$ to match the proper scales of the true velocity fields, and appropriately compute the wave equation 17.

**DPS** (Chung et al., 2023) From Bayes' rule, the posterior can be decomposed as

$$\nabla_{\boldsymbol{x}_t} \log p_t(\boldsymbol{x}_t | \boldsymbol{y}) = \nabla_{\boldsymbol{x}_t} \log p_t(\boldsymbol{x}_t) + \nabla_{\boldsymbol{x}_t} \log p_t(\boldsymbol{y} | \boldsymbol{x}_t), \tag{21}$$

where it is required to compute both the prior term $\nabla_{\boldsymbol{x}_t} \log p_t(\boldsymbol{x}_t)$, and the likelihood term $\nabla_{\boldsymbol{x}_t} \log p_t(\boldsymbol{y} | \boldsymbol{x}_t)$. The score function $\nabla_{\boldsymbol{x}_t} \log p_t(\boldsymbol{x}_t)$ can be obtained from pre-trained score networks $\boldsymbol{s}_{\boldsymbol{\theta}}(\boldsymbol{x}_t, t)$, and Gaussian approximation is used to compute the likelihood term, which results in a one-step gradient update

$$\nabla_{x_t} \log p_t(y | \boldsymbol{x}_t) \approx \rho \nabla_{x_t} \big\| y - \mathcal{A}(\hat{x}_0^{(t)}) \big\|_2^2, \tag{22}$$

where $\rho$ is the step size and $\hat{\boldsymbol{x}}_0^{(t)}$ is denoised estimate. For full waveform inversion problems in both CurveFault B and Style B datasets, the total number of time steps is set to $T = 1000$, with a step size of $\rho = 0.01$. For the Lorenz 1963 system, $T = 500$ with $\rho = 1$. For the Kolmogorov flow, $T = 1000$ with $\rho = 1$.

**DiffPIR** (Zhu et al., 2023) utilizes a variable splitting method to decouple the prior and likelihood steps. DiffPIR also employs a diffusion prior during the prior steps and solves the proximal subproblem

$$\hat{x}_0^{(t)} = \arg\min_{x} \| y - \mathcal{A}(x) \|^2 + \rho_t \| x - \hat{x}_0^{(t)} \|^2, \tag{23}$$

using a one-step gradient update for the likelihood steps with step size $\rho_t = \lambda(\sigma_n/\sigma_t)^2$,

$$\hat{x}_0^{(t)} \approx \hat{x}_0^{(t)} - \rho_t \nabla_{x_t} \| y - \mathcal{A}(\hat{x}_0^{(t)}) \|^2. \tag{24}$$

By choosing proper hyperparameters and schedules, DiffPIR aligns with the procedures of the DPS algorithm. However, our method uses multiple gradient updates starting from $\hat{x}_0^{(t)}$ (i.e., $\hat{x}_0^{(t)} = \arg\min_{x} \| y - \mathcal{A}(x) \|^2$ starting from denoised estimate $\hat{x}_0^{(t)}$ from previous step), instead of solving the proximal subproblem with one-step gradient update. Our method eliminates the need for cumbersome step-size tuning but adjusts the number of inner gradient updates per likelihood step for feasible time cost. The total number of time steps is set to $T = 1000$, with hyperparameters of regularization weight $\lambda = 10$, and stochasticity $\zeta = 0.1$.

**RED-diff** (Mardani et al., 2024) propose a variational approach leading to solving the following optimization problem at each noise level:

$$\mu = \arg\min_{x} \| y - \mathcal{A}(\mu) \|^2 + E_\epsilon \left[ \lambda_t \| \epsilon_\theta(x_t, t) - \epsilon \|^2 \right], \tag{25}$$

where $\lambda_t$ is the regularization weight which can be defined as a pre-defined schedule, they performed a single gradient update for the optimization process at each noise level, i.e., in a single reverse step, while our method utilizes multiple gradient updates for an optimization process at each reverse step. We implement the algorithm as one of our baselines. The total number of time steps is set to $T = 1000$, with linear schedule, and step size of the gradient updates is set to $0.01$, and regularization coefficients $\lambda = 0.1$.

**DAPS** (Zhang et al., 2024) also adopt a variable splitting method to decouple the prior and likelihood steps. Its inference procedure includes: (1) reverse process to obtain denoised estimate $\hat{x}_0^{(t)}$, (2) multiple number of gradient updates for a likelihood step, and (3) perturbation with the forward diffusion process to next noise level. While this approach is similar to ours, a key difference lies in their use of proximal loss term, $\|\hat{x}_0^{(j)} - \hat{x}_0\|^2$, during the likelihood gradient update. This term encourages solutions near the previous denoised estimate and follows the update rule:

$$\hat{x}_0^{(j+1)} = \hat{x}_0^{(j)} - \eta \nabla_{\hat{x}_0} \left( \frac{1}{2\beta_y^2} \|y - \mathcal{A}(x)\|^2 + \frac{1}{2r_t^2} \|\hat{x}_0^{(j)} - \hat{x}_0\|^2 \right), \tag{26}$$

where here is $\hat{x}_0$ denoised estimate of $x_t$ at time $t$. The total number of time steps is set to $T = 200$, and step size of the gradient updates is set to $\eta = 0.005$, and the number of inner updates 5.

**PCDM** Our method addresses the likelihood process by directly minimizing the term $\|y - \mathcal{A}(x)\|^2$ starting from the previous denoised estimate $\hat{x}_0^{(t)}$ as an initial guess,

$$\hat{x}_0^{(n+1)} = \hat{x}_0^{(n)} - \alpha \nabla_{\hat{x}_0^{(n)}} \|y - \mathcal{A}(\hat{x}_0^{(n)})\|_2^2, \qquad \hat{x}_0^{(0)} = z_{t_k}, \tag{27}$$

which does not require the hyperparameters of the weight coefficient $\beta_y$ and $\beta_t$ for the data-consistency term and proximal term. In our likelihood steps, we set a proper step size $\alpha$ and set the number of likelihood iterations $N$ for searching the solution near the previous state while strictly enforcing the physical constraints, rather than relying on balancing weights between measurement consistency and the proximal term. Moreover, our method conducts the likelihood steps to only the steps before $t < t_s$ during the reverse steps. By performing likelihood updates only in regions where they are more effective, we can achieve superior results within the same computational cost. For example, when 1000 likelihood iterations are conducted within a reverse process spanning $T = 200$, DAPS performs 5 inner gradient updates per likelihood step. In contrast, our method employs an unconditional reverse process for $t_s > 100$ and performs 10 inner gradient updates for $t_s < 100$. For full waveform inversion problems, the total number of time steps is set to $T = 200$, with step size $\alpha = 0.005$, the number of inner updates 10, and $ts/T = 0.5$. For the Lorenz 1963 system, the total number of time steps is set to $T = 100$, with step size $\alpha = 0.05$, the number of inner updates 10, and $ts/T = 0.5$. For the Kolmogorov flow, the total number of time steps is set to $T = 200$, with step size $\alpha = 0.05$, the number of inner updates 10, and $ts/T = 0.5$. For the topology optimization, the total number of time steps is set to $T = 200$, with step size $\alpha = 0.001$, the number of inner updates 10, and $ts/T = 0.5$. The overall algorithm of PCDM is presented in Algorithm 1, where we denote $x_{t_k}$ as $x_k$ for readability.

# B ADDITIONAL RESULTS

| Method | CurveFault B | | | | Style B | | | |
|---|---|---|---|---|---|---|---|---|
| | Res | MAE ↓ | RMSE ↓ | SSIM ↑ | Res | MAE ↓ | RMSE ↓ | SSIM ↑ |
| InversionNet | - | 1.67e-1 | 2.41e-1 | 0.605 | - | 5.86e-2 | 8.93e-2 | 0.760 |
| VelocityGAN | - | 1.58e-1 | 2.34e-1 | 0.603 | - | 6.49e-2 | 9.79e-2 | 0.725 |
| Opt w/o diff | 1.42e-3 | 2.17e-1 | 3.16e-1 | 0.410 | 2.53e-4 | 2.36e-1 | 3.25e-1 | 0.294 |
| DPS | 6.79e-4 | 1.29e-1 | 2.38e-1 | 0.632 | 1.49e-4 | 9.14e-2 | 1.35e-1 | 0.593 |
| Diffpir | 7.11e-4 | 1.15e-1 | 2.19e-1 | 0.670 | 1.96e-3 | 8.32e-2 | 1.24e-1 | 0.624 |
| RED-diff | 6.26e-4 | 8.43e-2 | 1.48e-1 | 0.751 | 8.60e-4 | 8.49e-2 | 1.26e-1 | 0.617 |
| DAPS | 4.12e-4 | 6.61e-2 | 1.19e-1 | 0.806 | 1.89e-4 | 3.79e-2 | 6.43e-2 | 0.837 |
| PCDM | **3.57e-5** | **4.89e-2** | **9.91e-2** | **0.850** | **4.96e-5** | **3.07e-2** | **5.37e-2** | **0.890** |

Table 5: Quantitative comparisons on full waveform inversion including state-of-the-art plug-and-play algorithms.

---

**Algorithm 1** Physics-constrained Diffusion Model (PCDM)

1: **Input:** Pre-trained diffusion model $\epsilon_\theta$, forward model $\mathcal{A}$, measurement $y$, time steps $\{t_0, \cdots, t_K\}$
2: **Hyperparameters:** Step size $\alpha$, number of iterations $N$, starting time of likelihood steps $t_s$
3: **Output:** $x_0$
4: Sample $x_T \sim \mathcal{N}(0, I)$
5: **for** $k = K, ..., 0$ **do**
6:     $z'_k \leftarrow \sqrt{\bar{\alpha}_k} x_{k+1} + \sqrt{1 - \bar{\alpha}_k} \hat{\epsilon}_{k+1}$         ▷ Prior step: DDIM sampling
7:     $x_k^{(0)} \leftarrow (z'_k - \sqrt{1 - \bar{\alpha}}_{t_k} \epsilon_\theta^{(k)}(z'_k))/\sqrt{\bar{\alpha}_k}$    ▷ Prior step: Obtaining denoised $x_k$
8:     **if** $t_k < t_s$ **then**                 ▷ Likelihood step: Starting time $t_s$
9:         **for** $n = 0, ..., N - 1$ **do**
10:            $x_k^{(n+1)} = x_k^{(n)} - \alpha \nabla_{x_k^{(n)}} \|y - \mathcal{A}(x_k^{(n)})\|_2^2$    ▷ Likelihood step: $N$ iterations
11:         **end for**
12:         $x_k \leftarrow x_k^{(N)}$
13:     **else**
14:         $x_k \leftarrow x_k^{(0)}$
15:     **end if**
16: **end for**
17: **return** $x_0$

---

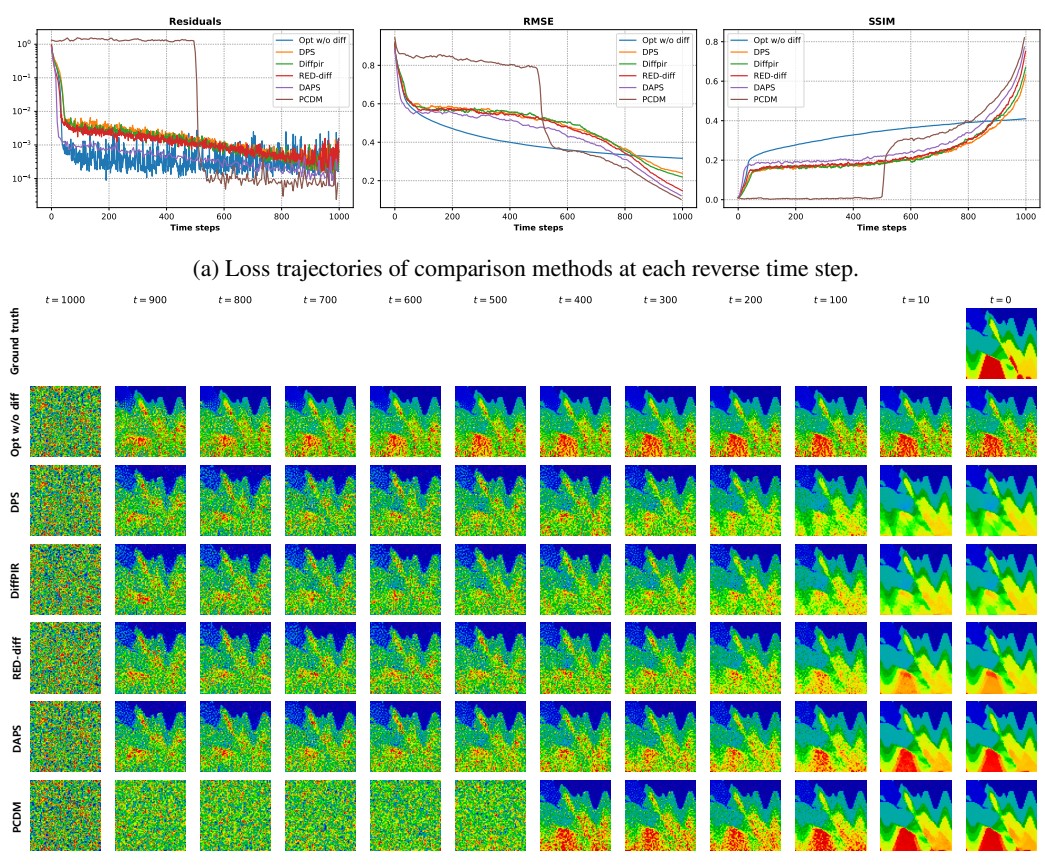

(a) Loss trajectories of comparison methods at each reverse time step.

(b) Progressive state at each reverse time step.

Figure 6: An example of (a) the loss trajectories of comparison methods and (b) the corresponding progressive states at each reverse time step, respectively (CurveFault B).

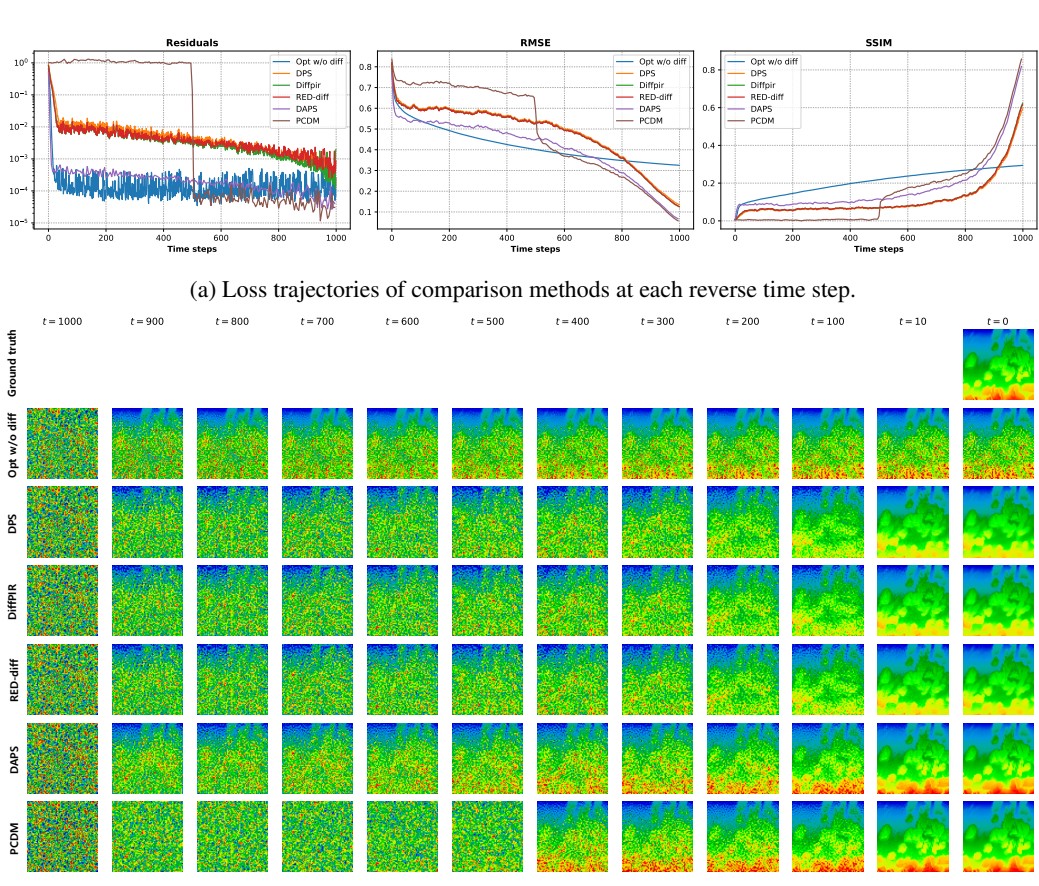

(a) Loss trajectories of comparison methods at each reverse time step.

(b) Progressive state at each reverse time step.

Figure 7: An example of (a) the loss trajectories of comparison methods and (b) the corresponding progressive states at each reverse time step, respectively (Style B).

