# OpenReview forum: "Efficient Physics-Constrained Diffusion Models for Solving Inverse Problems"
_ICLR.cc/2025/Conference — Submitted to ICLR 2025_

### Official Review · Reviewer_Liuf · 2024-10-23

**Soundness:** 1
**Presentation:** 1
**Contribution:** 1
**Rating:** 1
**Confidence:** 5

**Summary:**

The paper propose a method for the solution of an inverse problem where the forward problem is a solution of some physical simulating.
The claim is that the result of the algorithm produces solutions that do not only honour the prior but also obey the physics.

**Strengths:**

The results look interesting. The paper may be improved so the results will make sense.

**Weaknesses:**

To be honest I could not understand the paper, even though I have been working in this field for many years.  The authors invented new jargon, "physics constrained" which means, what exactly? What is the constraint they are fulfilling? On which variables? How do you deal with the constraints? Lagrange multipliers? elimination? penalty?
There is a huge branch of inverse problems that treat them as PDE constrained optimization. Clearly, this escaped from the authors. There is a large number of papers that introduce constraints into inverse problems (e.g 0 \le x) but clearly this is not one of these examples. The authors should try to rewrite the paper and be a bit more precise about what they do,

Similarly, in section 3, the equations flow smoothly and I can easily understand how to get from (7) and the way to (10).
Then you switch to section 3.2 and I cannot see how (11) and on is related to the previous section.

Finally in ADMM (eq 12) there is another term for Lagrange multiplier that you are missing. The solution of your problem is different than the original problem.

**Questions:**

1. What is physics constrained, please define mathematically

2. How to get from the Langevin dynamics of (8-9) to your optimization problem (11-12)

3. Why and how are the two related

4. Why don't you use Lagrange multipliers for (12), add a term p^T(z-x)

5. Can you clarify the overall algorithm?

6. How do you ensure that you fit the data to some given tolerance?

---

> ### Author Response · Authors · 2024-11-28
> **Response to Reviewer Liuf (part 1)**
>
> We appreciate the reviewer's insightful comments and constructive feedback. Our responses are given below:
>
> **The authors invented new jargon, "physics constrained" which means, what exactly? What is the constraint they are fulfilling? On which variables? How do you deal with the constraints? Lagrange multipliers? elimination? penalty?**
>
> The term “physics-constrained” is not new jargon. It has been used in scientific and engineering domains for many years [1, 2, 3, 4], where the solutions of the proposed methods are guided by underlying physical constraints or governing equations, aligning with our usage of the term. In our method, this is achieved by optimizing the penalty $\| y-A(x) \|_2^2$ during the diffusion reverse process, ensuring that the solution of the inverse problem $x$ is both physically plausible and adheres to the underlying physical constraints.
>
> [1] Physics-constrained deep learning for high-dimensional surrogate modeling and uncertainty quantification without labeled data, Journal of Computational Physics, 2019.
>
> [2] Surrogate Modeling for Fluid Flows Based on Physics-Constrained Deep Learning without Simulation Data, Computer Methods in Applied Mechanics and Engineering, 2020.
>
> [3] Learning physical models that can respect conservation laws, ICML, 2023.
>
> [4] multi-fidelity physics constrained neural networks for a dynamical system, Computer Methods in Applied Mechanics and Engineering, 2024.
>
> **There is a huge branch of inverse problems that treat them as PDE constrained optimization. ... but clearly this is not one of these examples.**
>
> We respectfully disagree with this statement. Our proposed method focuses on solving inverse problems in scientific and engineering domains, where the solution is obtained by optimizing a PDE-constrained objective with diffusion models as regularizers. This approach makes the solution more plausible and ensures it aligns with the underlying governing equation that the solution should satisfy.
>
> **Similarly, in section 3, the equations flow smoothly and I can easily understand how to get from (7) and the way to (10). Then you switch to section 3.2 and I cannot see how (11) and on is related to the previous section. / Question 2 / Question 3**
>
> **- Question 2: How to get from the Langevin dynamics of (8-9) to your optimization problem (11-12)**
>
> The Langevin dynamics described in equation (7-8) is a well-known method for solving the optimization problem (10-11). It leverages a generative process to progressively transition from $x_T \sim N(0, I)$ to the desired solution $x_0 \sim p(x|y)$. The sampling process during a small time step, transitioning from $t$ to $t-1$, is governed by (7), which requires the computation of the posterior score function $\nabla_{x_t} \log p_t (x_t|y)$. From Bayes’ rule, this score function can be decomposed into two terms: the score function $\nabla_{x_t} \log p_t(x_t)$, which can be computed using a pre-trained diffusion model with trainset, and the likelihood term, $\nabla_{x_t} \log p_t (y|x_t)$. [5] uses a Gaussian approximation for the likelihood function $\exp(-\rho \| y - A(x)\|^2 )$ which evaluates how well the solution satisfies the physical constraints. Correspondingly, the likelihood gradient term is approximately $\nabla_{x_t} \log p_t (y|x_t) \approx - \rho \nabla_{x_t} \| y – A(x) \|_2^2$.
>
> **- Question 3: Why and how are the two related**
>
> Roughly speaking, the sampling process from $t$ to $t-1$ can be interpreted as consisting of two steps: (1) a reverse step in the pre-trained (unconditional) diffusion generative process, transitioning from $x_t$ to $x_{t-1}$, and (2) a one-step gradient update to optimize the likelihood term $\min_x \frac{1}{2} \| y – A(x) \|_2^2$ as described in Equation (10). This can also be viewed as an optimizing equation (10), where the classical regularizer is replaced by the pre-trained diffusion reverse process.
>
> **- From equation (8,9) to equation (10, 11)**
>
> However, due to the one-step gradient update for the likelihood at every reverse time step – both $\nabla_{x_t} \log p_t (x_t)$ and $\nabla_{x_t} \log p_t(y|x_t)$ are computed simultaneously at every time t – this naïve approach can lead to slow inference times or suboptimal performance if the effective number of likelihood updates is insufficient to fully optimize the object. To address these limitations, it is necessary to employ accelerated diffusion sampling while enabling multiple gradient updates for the likelihood in an effective way. To this end, we revisit the original optimization problem (equation 10) and reformulate it inspired by variable splitting methods (equation 11). Such separation provides greater flexibility and efficiency, we can leverage accelerated sampling (DDIM) to reduce the number of reverse time steps. Moreover, we use multiple gradient updates within a single likelihood step and perform the updates concentrated on time steps where they are most effective.

---

> ### Author Response · Authors · 2024-11-28
> **Response to Reviewer Liuf (part 2)**
>
> **Question 1: What is physics constrained, please define mathematically**
>
> The goal of solving an inverse problem is to recover $x$ from the measurements $y$,
> $$y=A(x)+n,$$
> where A is the physical forward model such as PDEs and some physical constraints, and $n$ is additive noise. Then, the solution can be obtained by solving the following physics-constrained optimization problem,
> $$\min_x \frac{1}{2} \| y-A(x)\|_2^2 + \lambda R(x),$$
> where $L(x)=\frac{1}{2} \| y-A(x)\|_2^2$ is an objective function that stems from the likelihood of alignment for physics constraints. Therefore, the physics constraint means that solutions that are constrained by the objective $\frac{1}{2} \| y-A(x)\|_2^2$.
>
> **Finally in ADMM (eq 12) there is another term for Lagrange multiplier that you are missing. The solution of your problem is different than the original problem. / Question 4: Why don't you use Lagrange multipliers for (12), add a term p^T(z-x).**
>
> First, we reformulate the optimization problem with half quadratic splitting (HQS) methods which are usually used to solve the following form of optimization problem for the $x$,
> $$\min_x f(x)+g(x).$$
> This can be reformulated by
> $$\min_{x,z} f(x)+g(z)+\mu \| x -z \|^2.$$
>
> However, the alternating direction method of multipliers (ADMM), the reviewer mentioned, usually considers the following form of an optimization problem with two sets of variables,
> $$\min_{x,z} f(x)+g(z) \quad s.t. Ax+Bz=c,$$
> where the augmented Lagrangian for this problem is
> $$L_\rho(x, z, y) = f(x) + g(z) + y^T(Ax+bZ - c) + \rho \| Ax+Bz – c\|^2$$
> This is not consistent with our formulation, since one variable $x$ is optimized in our problem. Therefore, we respectfully disagree with the reviewer’s recommendation to add the term $\rho^T (z-x)$.
>
> **Question 5: Can you clarify the overall algorithm?**
>
> The overall algorithm of our method is presented in Algorithm 1 in Appendix A.3 (page 18).
> A noisy sample is drawn from a normal distribution, $x_T \sim N(0, I)$. During the prior step, we employ the DDIM sampling scheme, which transitions from time step $t_{k+1}$ to $t_{k}$ and obtains denoised estimate $x_k$, alternatively. If the current time step is less than $t_k < t_s$, we perform the $N$ gradient updates for likelihood steps. After completing the reverse processes, we obtain the solution $x_0$, which satisfies the given physical constraints.
>
> **Question 6: How do you ensure that you fit the data to some given tolerance?**
>
> Our inverse problems include highly nonlinear and complex forward models, where the optimization problem has no closed-form solution and it is not straightforward to obtain the theoretical error bounds of the solution for the optimization problem. Instead, we empirically demonstrate that our method outperforms existing state-of-the-art algorithms for solving inverse problems and we conducted thorough ablation studies on our hyperparameters including the step size of the optimizer, number of iterations within a likelihood step, and starting time of steps.

---

> > ### Comment · Reviewer_Liuf · 2024-11-28
> > **Retain my score**
> >
> > Not only that you did not answer my questions. Your answer shows deep gaps in understanding optimization theory.
> >
> > To start you clearly do not define constraints in a way that optimization theory does.  What does it mean "constrained by the objective $\frac{1}{2} | y-A(x)|_2^2$"? Typically one would formulate it as $\frac{1}{2} | y-A(x)|_2^2 \le epsilon$ or something like that. The misfit, $\frac{1}{2} | y-A(x)|_2^2$ is just a number, its not a constraint on $x$.
> > Also, $A$ is typically not a PDE! It is the solution operator. This is a huge difference. While a PDE is typically an unbounded operator the solution operator is typically compact. This is why inverse problems are ill-posed (rather than just ill-conditioned).
> >
> > Second, the claim that $$\min_x f(x)+g(x).$$  can be reformulated by $$\min_{x,z} f(x)+g(z)+\mu | x -z |^2.$$ is simply wrong!
> > I strongly suggest you try to do this for a simple problem and see what you get (even in 1D). You need the Lagrange multiplier to make them equivalent.
> >
> > Overall, this paper shows fundamental gaps in optimization theory and inverse problems. The authors should do their reputation good if they withdraw the paper and re-write what they want to say asking some advise from someone who is immersed in optimization.

---

### Official Review · Reviewer_DxjV · 2024-10-31

**Soundness:** 2
**Presentation:** 2
**Contribution:** 2
**Rating:** 5
**Confidence:** 4

**Summary:**

This paper proposes PCDM (physics-constrained diffusion model), an inverse problem solver that leverages diffusion model as plug-and-play prior. PCDM uses the idea of variable splitting and proposes to solve the underlying optimization problem with implicit diffusion model regularization. The authors demonstrate its application in full-waveform inversion, data assimilation, and topology optimization.

**Strengths:**

1. Applying plug-and-play diffusion model methods to physics-constrained inverse problems is relatively new to the diffusion model community.
2. The paper is generally easy to follow.

**Weaknesses:**

1. The proposed PCDM appears to be mathematically equivalent to a special case of the algorithm in Li et al. [1] (specifically, the case using Tweedie's formula). The claim of algorithmic novelty is questionable (line 100).
2. The "physics-constrained" aspect really comes from the inverse problem itself instead of the novel algorithmic design. Most existing gradient-based plug-and-play diffusion model methods can incorporate that physics loss such as DiffPIR [2], DPS,DAPS [3], RED-diff [4], [5]. These methods are not compared or  discussed in the paper.
3. The experimental comparison excludes many recent and relevant algorithms. For example DiffPIR [2] and DAPS [3], RED-diff [4].
4. Reproducibility concerns: important experimental and implementation details are insufficiently documented. See more concrete questions in the next section.
5. There is a lack of ablation studies on important algorithm design parameters, such as the number of likelihood steps per iteration, the optimization threshold $t_s$, and sensitivity to the optimizer configurations.


[1] : Li, Xiang, et al. "Decoupled data consistency with diffusion purification for image restoration." _arXiv preprint arXiv:2403.06054_ (2024).
[2] : Zhu, Yuanzhi, et al. "Denoising Diffusion Models for Plug-and-Play Image Restoration." _arXiv preprint arXiv:2305.08995_ (2023).
[3] : Zhang, Bingliang, et al. "Improving diffusion inverse problem solving with decoupled noise annealing." _arXiv preprint arXiv:2407.01521_ (2024).
[4] : Mardani, Morteza, et al. "A Variational Perspective on Solving Inverse Problems with Diffusion Models." _The Twelfth International Conference on Learning Representations_.
[5] : Peng, Xinyu, et al. "Improving Diffusion Models for Inverse Problems Using Optimal Posterior Covariance." _Forty-first International Conference on Machine Learning_. 2024.

**Questions:**

1. I'm a bit surprised at how well the Opt w/o diff baseline can recover the large structure of the ground truth, as shown in Figure 2 and Table 1. This contrasts with traditional FWI literature findings [1] and my own experimental validation on OpenFWI dataset. I'm curious how the authors implement the FWI problem and the corresponding baselines. More specifically,
	1. What is exactly the Opt w/o diff baseline in Table 1? Is that the Adam optimizer? What initialization strategy was employed? What are the specific hyperparameters used to report the results?
	3. Why are the residuals of InversionNet and VelocityGAN omitted from Table 1?
	4. Given that OpenFWI paper does not provide the gradient implementation of the forward model, how did the authors implement the gradient?
2. What are the hyperparameter selection criteria across compared methods?
3. Is there any supplementary material or code to facilitate the reproducibility?

[1] : Virieux, Jean, and Stéphane Operto. "An overview of full-waveform inversion in exploration geophysics." _Geophysics_ 74.6 (2009): WCC1-WCC26.

---

> ### Comment · Reviewer_DxjV · 2024-11-27
> **Individual response?**
>
> I noticed the short general response indicating that the paper has been revised. However, I did not see individual responses addressing the specific concerns raised by reviewers. Additionally, while I have checked the updated appendix and the revised paper, I could not easily locate the sections that address the major concerns outlined in my review.
>
> As a friendly reminder, the [ICLR policy](https://iclr.cc/Conferences/2025/AuthorGuide) states:
> > You can upload revisions until the discussion period ends, but reviewers and area chairs are not required to look at every revision. It is up to you to clearly communicate whats been changed.
>
> I would strongly encourage the authors to provide individual responses to each reviewer, explicitly stating how the revisions address the points raised. This would greatly facilitate understanding the changes made and their relevance to the feedback provided.

---

> ### Author Response · Authors · 2024-11-28
> **Response to Reviewer DxjV**
>
> Thank you for the notice. Additionally, we appreciate the reviewer's insightful comments and constructive feedback. Our responses are given below:
>
> **The proposed PCDM appears to be mathematically equivalent to a special case of the algorithm in Li et al. [1] (specifically, the case using Tweedie's formula). The claim of algorithmic novelty is questionable.**
>
> [1] also employs a variable splitting method and allows multiple gradient updates within a likelihood step, similar to DAPS [3]. The key algorithmic difference between our method and [1, 3] is that our approach begins performing likelihood steps after $t < t_s$ (e.g., $t_s/T=0.25$) of the reverse process. This reduces the round of likelihood step overall, allowing us to perform more gradient updates in the selective time steps ($t_s<t$) given the same computational resources. In Appendix A.3, we describe the difference between DAPS [3] and our method and present thorough comparisons with both quantitative (Table 5) and qualitative (Tables 6 and 7) results. These experiments demonstrate that performing more likelihood gradient updates in the later stages of the reverse process is more effective than applying the same number of updates across all reverse steps.
>
> [1] Decoupled data consistency with diffusion purification for image restoration, arXiv, 2024.
>
> [3] Improving diffusion inverse problem solving with decoupled noise annealing, arXiv, 2024.
>
> **The "physics-constrained" ... These methods are not compared or discussed in the paper. / The experimental comparison excludes many recent and relevant algorithms. For example DiffPIR and DAPS, RED-diff.**
>
> As noted by the reviewer, we have provided a brief explanation of the state-of-the-art methods (including DPS, DiffPIR, RED-diff, and DAPS) and discussed their difference from our approach in Appendix A.3. Additionally, we conducted additional thorough experiments comparing these state-of-the-art algorithms with our method in Table 5, and Figure 6, 7 in Appendix B. Our proposed method outperforms the state-of-the-art comparisons in all evaluation metrics.
>
> **What is exactly the Opt w/o diff baseline in Table 1? Is that the Adam optimizer? What initialization strategy was employed? What are the specific hyperparameters used to report the results?**
>
> We employ the Adam optimizer with a learning rate of 0.005 and perform 1,000 iterations for the total optimization process. For the initialization, we take a random initialization from a standard normal distribution $N(0, I)$. During the optimization, $x$ is scaled to match the proper scales of the values of velocity fields (about 1500 - 4500 m/s). We calculate the minimum and maximum values, $v_{min}$, and $v_{max}$ from the training set of velocity fields, which are used for denormalization.
>
> **Why are the residuals of InversionNet and VelocityGAN omitted from Table 1?**
>
> InversionNet and VelocityGAN are examples of the end-to-end method which don’t include the forward model therefore, it cannot compute the residual of measurement-consistency term $\| y-A(x) \|$.
>
> **Given that OpenFWI paper does not provide the gradient implementation of the forward model, how did the authors implement the gradient?**
>
> We utilized the open-sourced Deepwave package [1], which implements the forward model using PyTorch. To compute the gradient and optimize the process on the target velocity field, we employed torch.autograd.grad and torch.optim.Adam.
>
> [1] Richardson, A. (2023). Deepwave (v0.0.20). Zenodo. https://doi.org/10.5281/zenodo.8381177
>
> **What are the hyperparameter selection criteria across compared methods?**
>
> The hyperparameters for the baselines are provided in Appendix A.3, and implementation details of problems and training are provided in Appendix A.
>
> **Is there any supplementary material or code to facilitate the reproducibility?**
>
> We will release our implementation code following the publication.
>
> **There is a lack of ablation studies on important algorithm design parameters, such as the number of likelihood steps per iteration, the optimization threshold t_s, and sensitivity to the optimizer configurations.**
>
> As noted by the reviewer, ablation studies on key hypermeters and their effectiveness are presented in Figure 5 in Section 4.4 Ablation studies. The figures highlight that selecting an appropriate step size (such as $\alpha=5e-3$ in that case) is essential. Performing more likelihood iterations per likelihood step leads to better performance. Furthermore, performing likelihood steps only after $t < t_s$ (e.g., $t_s/T=0.25$) of the reverse process achieves comparable results with significantly reduced computational time. These findings highlight the flexibility and efficiency of our algorithm in addressing inverse problems within scientific domains, making it a practical use.

---

> > ### Comment · Reviewer_DxjV · 2024-12-03
> >
> > Thank the authors for posting individual responses. The rebuttal helps clarify several points but the novelty concern still remains unaddressed.
> >
> > **Novelty** If the primary difference is the starting iteration of the likelihood step, I don't think the proposed PCDM can be considered a novel algorithm. While this detail may have some impact on computational efficiency, it is unclear if this constitutes sufficient algorithmic novelty for ICLR publication. I also note that the current manuscript does not give adequate credit to [1] despite its apparent influence on your proposed method.
> >
> > Furthermore, I'm not sure there is enough new knowledge or sufficient value for the community. The only thing I learned from the paper is that the authors show that an existing algorithm works well on a set of known and curated toy problems different from the common image restoration tasks, which does not provide sufficient new insights.
> >
> > A stronger contribution would involve addressing previously unresolved challenges or introducing new ideas and insights.
> >
> >
> > **Residual of InversionNet and VelocityGAN** While I understand that InversionNet and VelocityGAN are end-to-end networks that do not incorporate the forward model explicitly, these methods still produce predictions $\hat{x}$. Using the Deepwave implementation, it should be possible to compute the residual $\|y-A(\hat{x})\|$, enabling a fair comparison of measurement consistency across different methods.
> >
> > **hyperparameter selection criteria** Appendix A.3 reports the hyperparameter choices but does not provide insight into how these values were selected or tuned. My question pertains to the criteria and process used for selecting hyperparameters across the compared methods. Directly borrowing hyperparameters from prior work without adaptation seems inappropriate, especially since your experimental setups differ significantly from the original papers.
> >
> > I appreciate the authors’ efforts in responding to the reviewers and revising the manuscript, but based on the points above, I remain concerned about the degree of novelty and clarity in key experimental details.
> >
> > [1]: Decoupled data consistency with diffusion purification for image restoration, arXiv, 2024.

---

### Official Review · Reviewer_RYMp · 2024-10-31

**Soundness:** 2
**Presentation:** 2
**Contribution:** 1
**Rating:** 3
**Confidence:** 4

**Summary:**

The paper proposes doing MAP estimation using a diffusion plug and play prior (PnP). Namely, the paper aims at solving
$$argmin \|y - \mathcal{A}(x)\| + \lambda \mathcal{R}(x).$$

To do so, it follows the traditional PnP route by using ADMM to split this into solving two proximal problems:

$$ z_{i+1} = argmin_{z} \mathcal{L}_\mu(z, x_i) $$

$$ x_{i+1} = argmin_{x} \mathcal{L}_\mu(z_{i+1}, x) $$

Finally, it replaces the prior proximal problem by a forward backward (with one step) sampling, namely equation (15).
It then evaluates the approach in non-linear problems coming from physics.

**Strengths:**

The paper proposes an adapted method to solving several physics problems where adapting to a physical constraint is cast as having high likelihood. The numerical applications are relevant.

**Weaknesses:**

My main concern is the novelty aspect of the paper. Indeed, several papers have investigated the applications of pretrained diffusion generative models as PnP priors. In particular, Algorithm 1 of [1] is essentially the same as the one proposed in this paper. Unless I'm mistaken, this makes the only novelty in this paper w.r.t. [1] to be the physical applications, which are indeed interesting. But I do not reckon it is worth being accepted to ICLR.

Furthermore, even if the proposed algorithm is conceptually different, it is still part of the broad Plug and Play family and I would expect at least a comparison with [1] or any other Plug and Play with diffusion paper.


[1]  Denoising Diffusion Models for Plug-and-Play Image Restoration
Yuanzhi Zhu, Kai Zhang, Jingyun Liang, Jiezhang Cao, Bihan Wen, Radu Timofte, Luc Van Gool; Proceedings of the IEEE/CVF Conference on Computer Vision and Pattern Recognition (CVPR) Workshops, 2023, pp. 1219-1229

[2] Provably robust score-based diffusion posterior sampling for plug-and-play image reconstruction.
 Xu, Xingyu, and Yuejie Chi. arXiv preprint arXiv:2403.17042 (2024).

[3]Graikos, Alexandros, et al. "Diffusion models as plug-and-play priors." Advances in Neural Information Processing Systems 35 (2022): 14715-14728.

[4] F. Coeurdoux, N. Dobigeon and P. Chainais, "Plug-and-Play Split Gibbs Sampler: Embedding Deep Generative Priors in Bayesian Inference," in IEEE Transactions on Image Processing, vol. 33, pp. 3496-3507, 2024, doi: 10.1109/TIP.2024.3404338.

[5] Wu, Zihui, et al. "Principled Probabilistic Imaging using Diffusion Models as Plug-and-Play Priors." arXiv preprint arXiv:2405.18782 (2024).

[6] Wang, Hengkang, et al. "DMPlug: A Plug-in Method for Solving Inverse Problems with Diffusion Models." arXiv preprint arXiv:2405.16749 (2024).

**Questions:**

For the major point, see weaknesses.

Minor questions and remarks.

* Is the left term in eq(6) $x_{t-1}$ ? Otherwise it is not a sampling process, as it does not evolve through time.
* What is $ \hat{\epsilon}_{t}$ in equation (15) ?
Is it equation (7) with $x_{t_k}$?
* Equation (15) mixes indexes between $t_k$ and $t$.

---

> ### Author Response · Authors · 2024-11-28
> **Response to Reviewer RYMp**
>
> We appreciate the reviewer's insightful comments and constructive feedback. Our responses are given below:
>
> **My main concern is the novelty aspect of the paper. Indeed, several papers have investigated the applications of pretrained diffusion generative models as PnP priors. Algorithm 1 of [1] is essentially the same as the one proposed in this paper. Unless I'm mistaken, this makes the only novelty in this paper w.r.t. [1] to be the physical applications, which are indeed interesting.**
>
> Applications to scientific and engineering domains are not trivial. Existing state-of-the-art plug-and-play algorithms are primarily applied in image restoration tasks, where the forward models are typically degradation operations represented by linear matrices or convolutional operators, which are less expensive and relatively less complex. In contrast, solving inverse problems in scientific and engineering domains involves forward models based on physical simulations or partial differential equations, which are significantly more computationally intensive and complex.
>
> The key difference between [1] and our proposed method lies in how likelihood steps are performed. While [1] employs a single gradient update for the likelihood steps, our method introduces the flexibility of performing multiple gradient updates ($N$) and applies the likelihood steps selectively, focusing only on the later steps of the diffusion reverse process ($t<t_s$). Our observations indicate that likelihood steps have minimal impact during the early stages but become more effective later in the process. This design choice improves both efficiency and performance.
>
> [1] Denoising Diffusion Models for Plug-and-Play Image Restoration CVPRW, 2023.
>
> **Furthermore, ... I would expect at least a comparison with [1] or any other Plug and Play with diffusion paper.**
>
> As noted by the reviewer, we have provided a brief explanation of the state-of-the-art methods including [1], and discussed their difference from our approach in Appendix A.3. Additionally, we conducted additional thorough experiments comparing these state-of-the-art algorithms with our method in Table 5, and Figure 6, 7 in Appendix B. Our proposed method outperforms the state-of-the-art comparisons in all evaluation metrics.
>
> **Is the left term in Eq 6 x_(t-1)? Otherwise, it is not a sampling process, as it does not evolve through time. / What is $ \hat{\epsilon}{t}$ in Eq 15? Is it Eq 7 with $x_{t_k}$? / Equation 15 mixes indexes between $t_k$ and $t$.**
>
> Yes, the term should be $x_{t-1}$ / Yes, that is the same with $\hat{\epsilon_t}$ (previously Eq 7).
> The term $\hat{\epsilon_t}$ is the noise term in the DDIM sampling process which is a weighted combination of deterministic $\epsilon_\theta^{(t)} (x_t)$ and stochastic $\epsilon \sim N(0, I)$ component. For readability, we denote $x_{t_k}$ as $x_k$ in the revised manuscript to address the issue of mixing indexes. I appreciate the reviewer pointing out the typo.

---

### Official Review · Reviewer_uCwN · 2024-10-31

**Soundness:** 3
**Presentation:** 2
**Contribution:** 2
**Rating:** 6
**Confidence:** 4

**Summary:**

This paper proposes a variable splitting method for solving physics-constrained inverse problems with diffusion model priors. Throughout the reverse diffusion process, the method alternates between two optimization problems: one to update the noisy estimated image with the diffusion model as a regularizer, and one to enforce data/physics constraints. The authors present experiments on full-waveform inversion, data assimilation, and topology optimization, showing quantitative and qualitative improvement upon baselines in all three applications.

**Strengths:**

* The proposed method is simple and intuitive. Although the methodology lacks technical novelty (see weaknesses), it’s helpful to see that such a simple extension of DPS and other plug-and-play diffusion-based inverse solvers may already go a long way in handling physics inverse problems.
* Validation is done on three very different tasks, and task-specific baselines and the DPS baseline are compared against for each task.

**Weaknesses:**

* The technical contribution is marginal. The idea of variable splitting for diffusion-based inverse solving is not new (Equation 16 is similar to the proximal optimization step in Equation 8 of Song and Shen et al. 2022). The main difference in this work is that there may not be a closed-form solution to Equation 16, so iterative gradient-based optimization is used at each likelihood step.
* In Tables 1 and 2, the smaller number of reverse steps used by PCDM is touted. This is a little misleading, as the Table 1 caption says that PCDM involves 1000 likelihood iterations in addition to 200 reverse steps. For expensive forward models, it may be the case that these 1000 likelihood iterations are very costly. Also, a clarifying question: does one likelihood iteration mean an entire optimization round of Equation 16, or do the 1000 likelihood iterations account for all the gradient steps needed to solve Equation 16 throughout the algorithm?
* I have concerns about how fair the comparison to DPS is. In Figure 3(b), the DPS reconstruction clearly doesn’t match the visual statistics of Kolmogorov flow. I would expect DPS to at least produce something that appears visually plausible even if it doesn’t agree with the physical model. For example, in Figure 4 of SDA (Rozet and Louppe 2023) and Figure 5 of Feng et al. 2024, the DPS reconstructions at least look qualitatively reasonable. I would also be curious how hyperparameters for DPS were chosen.

---
References:

Song and Shen et al. “Solving Inverse Problems in Medical Imaging with Score-Based Generative Models.” ICLR 2022.

Rozet and Loupe. “Score-based Data Assimilation.” NeurIPS 2023.

Feng et al. “Neural Approximate Mirror Maps for Constrained Diffusion Models.” arXiv 2024.

**Questions:**

* Please comment on how hyperparameters for baselines, including DPS, were chosen. I recommend making an appendix to include such details.
* Often it makes more sense to think of physics constraints as priors (i.e., checking whether a solution satisfies a physical model doesn’t involve the observed measurements). Does it make sense in that case to move the physics-consistency term into Equation 12 as an additional regularizer?
* It’s surprising that “Opt w/o diff” in Table 1 has the worst data residual, given that it only optimizes the likelihood term. The authors suggest that this is because it struggles with local minima, but I was under the impression that adding a diffusion regularizer would only complicate the optimization landscape. I would appreciate comments from the authors on why they believe “opt w/o diff” struggles to fit the data and whether they observed the same trend with the other tasks (why wasn’t opt w/o diff included as a baseline for the other tasks?).

---

> ### Author Response · Authors · 2024-11-28
> **Response to Reviewer uCwN (part 1)**
>
> We appreciate the reviewer's insightful comments and constructive feedback. Our responses are given below:
>
> **The technical contribution is marginal. The idea of variable splitting ... likelihood step.**
>
> As noted by the reviewer, the main difference is that there is no closed-form solution for our optimization problem. Therefore, we used a gradient update for the likelihood is used for each likelihood step. However, another key contribution of our method lies in how the likelihood steps are performed. While existing state-of-the-art plug-and-play (PnP) algorithms also rely on gradient updates for the likelihood, our approach has greater flexibility and efficiency. Unlike existing PnP algorithms, which use a single gradient update per likelihood steps, our method introduces the flexibility of performing multiple gradient updates ($N$) and applies the likelihood steps selectively, focusing on the later steps of the diffusion reverse process ($t<t_s$). Our observations indicate that likelihood steps have minimal impact during the early stages but become more effective later in the process. This design choice improves both efficiency and performance, as described in Table 5, and Figure 6 and 7, with thorough comparisons.
>
> **Does one likelihood iteration mean ... throughout the algorithm?**
>
> The 1000 likelihood iterations represent the total gradient updates required to solve the inverse problem, rather than the updates needed for a single round of Equation 15 (previously Equation 16). To ensure a fair comparison, both DPS (1000) and PCDM (200) use an equal number of likelihood iterations. However, DPS involves an additional 1000 reverse diffusion steps, whereas PCDM incorporates 200 reverse diffusion steps. To clarify, we present the gradient update scheme in Equation 16, which corresponds to a single likelihood iteration.
>
> **Concerns about how fair the comparison to DPS is, such as in Figure 3 (b). / Please comment on how hyperparameters for baselines, including DPS with an appendix.**
>
> We used the same setting of pre-trained diffusion model and hyperparameter settings as closely as possible to the implementation in [1] for all comparison methods. Although minor differences exist, such as preprocessing or visualized examples, the comparison between DPS, SDA, and PCDM remains fair, as they all utilize the same pre-trained diffusion model, during inference. As recommended by the reviewer, we added details on the training process and used hyperparameters in Table 4 and Appendix A.2 and 3.
>
> [1] Score-based Data Assimilation, Neurips 2023.
>
> **Often it makes more sense to think of physics constraints as prior ... move the physics-consistency term as an additional regularizer?**
>
> Our framework treats both physical model and measurement operators (represented by sparse matrix or convolution operators with a given kernel) as the forward model $A(x)$, and the corresponding likelihood term, $\| y – A(x) \|_2^2$, is treated as the same way.
> For example, in the case of our data assimilation scenarios, we consider two types of constraints including sparse measurements, represented by $y_1=M(x)$, where $M$ is a forward model with 8x spatial coarsening and 4x temporal coarsening operation and physical residuals, represented as $y_2 = r = P(x)$, where $P(x)$ represents the governing equation (e.g., $P(x) = 0$). Therefore, the corresponding likelihood terms can be a combination of them, $c_1 \cdot \| y_1 – M(x) \|_2^2 + c_2 \cdot \| 0 – P(x) \|_2^2$.
> From the domain-specific problem perspective, physics constraints are often the primary target for minimization to obtain a physically plausible solution. This aligns with the reviewers’ question to consider the physics-consistency term as prior. However, in our approach, the obtaining solution to the inverse problem is formulated as a sampling process from a pre-trained generative model. This process is iteratively guided by alignment with either observations or physics-based constraints, where these constraints are enforced as likelihood rather than treated as prior.

---

> ### Author Response · Authors · 2024-11-28
> **Response to Reviewer uCwN (part 2)**
>
> **I would appreciate comments from the authors on why they believe “opt w/o diff” struggles to fit the data and whether they observed the same trend with the other tasks (why wasn’t opt w/o diff included as a baseline for the other tasks?).**
>
> To provide strong evidence, we included loss trajectories (blue lines represent Opt w/o diff) and progressive states at each time step, as shown in Figures 6 and 7 (in Appendix B), to illustrate the optimization landscape. As these figures demonstrate, the only optimizing the likelihood term struggles with local minima, leading to poor performance despite rapid initial convergence. In contrast, the comparisons incorporating the pre-trained diffusion show a poor start at the early stage, and they finally show better results. These models, with their ability to capture the data structure of velocity fields, can generate the solutions with velocity fields-like images. Given the ill-posed nature of inverse problems, the usage of an appropriate regularizer is often crucial to obtaining plausible solutions. Diffusion models, with their expressive capacity to capture complex data structures, serve as powerful regularizers with their iterative generative process.
>
> In the case of topology optimization, the SIMP method (based on the finite element method, which typically takes a long time to converge) is used in Figure 4 and Table 3 to present the optimized solution without diffusion. For this task, the training set of optimal topologies generated by the SIMP is used to train the diffusion model. The evaluation metric "% CE" indicates the stability of the structure relative to the SIMP solution, which is a common metric in the related literature [1, 2]. The negative values in Figure 4 and Table 3 indicate that our method, which combines optimization with the diffusion model, generates more stable structures under the given boundary conditions.
>
> [1] Diffusion Models Beat GANs on Topology Optimization, AAAI, 2023.
>
> [2] Aligning Optimization Trajectories with Diffusion Models for Constrained Design Generation, NeurIPS, 2023.

---

### Official Review · Reviewer_LNSc · 2024-11-04

**Soundness:** 1
**Presentation:** 1
**Contribution:** 2
**Rating:** 3
**Confidence:** 5

**Summary:**

The authors address inverse problems by leveraging diffusion models as priors.
They formulate the problem as minimizing a composite objective function comprising a likelihood term, which enforces physics constraints, and a regularization term defined by a diffusion model.
As the resulting problem is difficult to solve directly, the authors utilizes a variable splitting scheme that alternates between minimization over the regularizer and the likelihood.
The regularization step is handled through a backward diffusion step, while the likelihood step is performed by minimizing and L2-regularized inverse problem.
The authors validate their approach on a set of three problems.

**Strengths:**

Solve inverse problems that arise in physics-constrained setups using a variable splitting scheme.

**Weaknesses:**

**Insufficient coverage of the related work**

The authors provide a high-level overview of two lines of research: end-to-end supervised approaches and unsupervised approaches.
While, there is a wealth of methods in inverse problems with diffusion models priors, few are mentioned.
Notably, related works that leverage variable splitting schemes, also known as Split Gibbs sampling, are not discussed; for reference, see [1, 2, 3] and the corresponding Related Work sections.

**Methodological ambiguities**

Section 3.3 introduces the regularization step without a clear justification for its formulation. Specifically, _why it has this form?_.
Furthermore, the method employs two regularization hyperparameters, $\lambda$ and $\mu$, yet only $\mu$ appears in the update equations.
Besides, the regularization step is independent of these hyperparameters.

**Lack of implementation details**

- The paper does not address the sensitivity of the method to its hyperparameters, namely the early stopping criterion and the timing of triggering the optimization (the parameter $t_s$ in line 256).
- The experimental section lacks specific implementation details, such as the hyperparameters for DPS and SDA; details regarding the used pre-trained diffusion models.
- The reported results raises some concerns In Table 1, DPS performance appears almost identical to the method that omits the prior (Opt w/o diff), which warrants further clarification as inverse problems are severely ill-posed hence pure optimization often yields an inconsistent solutions

---
.. [1] Zhu, Yuanzhi, et al. "Denoising diffusion models for plug-and-play image restoration." Proceedings of the IEEE/CVF Conference on Computer Vision and Pattern Recognition. 2023.

.. [2] Wu, Zihui, et al. "Principled Probabilistic Imaging using Diffusion Models as Plug-and-Play Priors." arXiv preprint arXiv:2405.18782 (2024).

.. [3] Xu, Xingyu, and Yuejie Chi. "Provably robust score-based diffusion posterior sampling for plug-and-play image reconstruction." arXiv preprint arXiv:2403.17042 (2024).

.. [4] Rozet, François, and Gilles Louppe. "Score-based data assimilation." Advances in Neural Information Processing Systems 36 (2023): 40521-40541.

**Questions:**

**Specific questions**

In the experiments, the formulation of the inverse problem in Experiments 4.1 and 4.3 is unclear, namely

- Experiment 4.1: is the operator $A$ a discretization of the d’Alembert operator? Additionally, is $s(r,t)$ provided within the dataset?
 - Experiment 4.3: Given that the problem is defined as a constrained optimization, how does the operator $A$ transform $x$ to yield the observation $y$?

Why was SDA excluded from Experiments 4.1 and 4.3? Although originally developed for data assimilation, it remains applicable as an inverse problem method.
Similarly, why was DPS omitted from Experiment 4.3?


**Broader questions**

- Could this method be applied to inverse problems in image restoration, and how would it compare to existing algorithms in the literature?
- The paper addresses problems of moderate dimensionality, approximately $5000$; have the authors considered Sequential Monte Carlo methods [1, 2, 3], which offer stronger theoretical guarantees?
Given this dimensionality, propagating multiple particles in parallel is feasible and would overcome mode-collapse.

---
.. [1] Dou, Zehao, and Yang Song. "Diffusion posterior sampling for linear inverse problem solving: A filtering perspective." The Twelfth International Conference on Learning Representations. 2024.

.. [2] Cardoso, Gabriel, Janati Yazid,, Sylvain Le Corff, and Eric Moulines. "Monte Carlo guided Denoising Diffusion models for Bayesian linear inverse problems." The Twelfth International Conference on Learning Representations. 2023.

.. [3] Wu, Luhuan, et al. "Practical and asymptotically exact conditional sampling in diffusion models." Advances in Neural Information Processing Systems 36 (2024).

---

> ### Author Response · Authors · 2024-11-28
> **Response to Reviewer LNSc (part 1)**
>
> We appreciate the reviewer's insightful comments and constructive feedback. Our responses are given below:
>
> **Insufficient coverage of the related work**
>
> As noted by the reviewer, we have provided a brief explanation of the state-of-the-art methods and discussed their difference from our approach in Appendix A.3. Additionally, we conducted additional thorough experiments comparing these state-of-the-art algorithms with our method in Table 5, and Figure 6, 7 in Appendix B.
>
> **Methodological ambiguities**
>
> The regularization step corresponds to the DDIM sampling scheme, as described in Equation (6) of Section 3.1. By utilizing DDIM, we accelerate reverse sampling with fewer steps.
> The $\lambda$ serves as weight coefficients between regularizer $R(z)$ and proximal term $\| z- x_k \|$, and $\mu$ serves as weight coefficient between measurement-consistency term $\|y-A(x)\|$ and proximal term $\|z – x_k\|$. Instead of explicitly tuning these hyperparameters, we implicitly implement the proximal operator $\|z-x_k\|$ through sampling or optimizing starting from the output of the previous steps. During the regularizer steps, sufficiently small changes between $t_k$ and $t_{k-1}$ ensure that the state of the next time step, regularized by the diffusion model, remains close to the state from the previous step. In our likelihood steps, we set a proper step size $\alpha$ and limit the number of likelihood updates $N$ for searching the solution near the previous state while strictly enforcing the physical constraints, rather than relying on balancing weights between measurement consistency and the proximal term. This approach removes the need for the $\mu$ and $\lambda$ by implicitly alternating between reverse sampling and optimization, with each process initialized using the output of the previous steps. Empirical studies validating the effectiveness of our hyperparameters are provided in Figure 5 in Section 4.4 Ablation studies.
>
> **Lack of implementation details**
>
> We included ablation studies in Figure 5 and Section 4.4. Additionally, details of the architectures training procedures, and implementation details for both our methods and the baselines are provided in Appendix A.
>
> **Specific questions**
>
> **Experiment 4.1 ... dataset?**
>
> Roughly speaking, the velocity field is represented as $x = v(r)$, and the seismic measurement is represented as $y= p(r,t)$. The wave equation is expressed as $A(x)y = s$, where $A$ incorporates the Laplace operator and second-order time derivatives. Consequently, the solution is given by $y=A^{-1}(x)s$. This formulation is implemented using the finite difference method. For the source function $s(r, t)$, the locations and waveform of source functions are predefined in the benchmark paper [1] and further details are described in Appendix A.1 Problem details.
>
> [1] OpenFWI: Large-scale Multi-structural Benchmark Datasets for Full Waveform Inversion, NeuriPS 2022.
>
> **Experiment 4.3 ... observation?**
>
> Topology optimization includes three constraints; compliance $C(x)=U^T (x) K^T (x)U(x)$ near to zero, elastic equilibrium $K(x)U(x)=F$, and volume constraint, $\frac{1}{N}\sum_i x_i -V_0 \leq 0$, where $K(x)$ and $U(x)$ are the global stiffness and displacement respectively, and $F$ is given loads. Therefore, we implement the constraint optimization problem as, $\underset{x}{argmin}  \| K(x)U(x) - F\|_2^2 + c_1 \cdot \| \mathcal{C}(x) - 0\|_2^2 + c_2 \cdot ReLU(\frac{1}{N}\sum_i x_i - V_0)$, where given loads and volume conditions can be considered as observations and the compliance and elastic equations can be considered as the forward operator. We set the coefficients with $c_1=1e-4$ and $c_2=1$.
>
> **Why was SDA excluded from Experiments 4.1 and 4.3?**
>
> Different from existing data assimilation methods, which restore each frame $x_i$ solely from the incomplete observation of itself, SDA uses surrounding frames $x_{i-k:i+k}$ within a window size $2k$ to restore the target frame $x_i$. While this approach is well-suited for sequential data, it is not directly applicable to other inverse problems that require reconstructing a single target frame $x$.
>
> **Broader questions**
>
> **Could this method be applied to inverse problems in image restoration?**
>
> Yes, this algorithm is compatible with image restoration. However, in this paper, we focused on the scientific and engineering domains. While some physical problems could be considered as image restoration from sparse measurements, exploring such applications is outside the scope of our interest, as we aim to address problems that incorporate physical simulations or constraints.

---

> ### Author Response · Authors · 2024-11-28
> **Response to Reviewer LNSc (part 2)**
>
> **The paper ... Have the authors considered Sequential Monte Carlo methods [1, 2, 3], ... mode-collapse.**
>
> Sequential Monte Carlo methods mentioned by the reviewer are usually applied to linear inverse problems. In our work, the forward models involve complex nonlinear operators, which pose significant challenges for the direct application of these methods. Although these methods have strong theoretical guarantees, further development would be required to adapt them effectively to inverse problems in scientific and engineering domains.

---

### Comment · Area_Chair_JVki · 2024-11-25
**Authors' Rebuttal**

Dear Authors,

As the author-reviewer discussion period is approaching its end, I strongly encourage you to read the reviews and engage with the reviewers to ensure the message of your paper has been appropriately conveyed and any outstanding questions have been resolved.

This is a crucial step, as it ensures that both reviewers and authors are on the same page regarding the paper's strengths and areas for improvement.

Thank you again for your submission.

Best regards,

AC

---

### Author Response · Authors · 2024-11-26
**General response**

We would like to appreciate all the reviewers for their constructive comments which have led to the revision, and we believe without a doubt have improved the quality of the manuscript. The revised manuscript has been uploaded. We have addressed the individual reviewer's comments below respectively, and here the summary of the major changes is as follows:

1.	We added ablation studies in Figure 5 and Section 4.4 in page 10.

2.	We added details of the architectures and training procedures, and implementation details for our methods in Appendix A.1 and 2.

3.	We have provided a brief explanation of the state-of-the-art methods and discussed their difference from our approach in Appendix A.3. Additionally, we conducted additional experiments comparing these state-of-the-art algorithms with our method in Table 5, and Figure 6, 7 in Appendix B.

4. We added the overall algorithm of our method in Algorithm 1 (Appendix A.3).

---

### Meta-Review · Area_Chair_JVki · 2024-12-20

**Metareview:**

This paper introduces Physics-Constrained Diffusion Model (PCDM), for solving inverse problems in physics by leveraging diffusion models as priors. PCDM employs a variable splitting technique, similar to ADMM and plug-and-play methods, to minimize a composite objective function that balances a likelihood term (enforcing physical constraints) with a regularization term defined by a diffusion model. This is achieved by alternating between a step that updates the solution using the diffusion model as a regularizer and a step that enforces data and physics constraints. The authors demonstrate PCDM's effectiveness on three applications: full-waveform inversion, data assimilation, and topology optimization, showing improved results compared to baseline methods. The claimed contribution lies in the integration of diffusion model priors with physical constraints to achieve solutions that are both realistic and physically consistent.

All the reviewers agree that the novelty of this approach is limited, and that the authors seem oblivious to the large amount of literature in the topic. As such the benchmarks are not really useful as they are not compared with state-of-the-art related (or in fact very similar) methods. As such I recommend rejection.

**Additional Comments On Reviewer Discussion:**

Most of the reviewers agree with the lack of novelty. The response of the authors didn't address their concerns.

---

### Decision · Program_Chairs · 2025-01-22

Reject